# What Is Fusarium Head Blight (FHB) Resistance and What Are Its Food Safety Risks in Wheat? Problems and Solutions—A Review

**DOI:** 10.3390/toxins16010031

**Published:** 2024-01-08

**Authors:** Akos Mesterhazy

**Affiliations:** Cereal Research Non-Profit Ltd., Alsokikotosor 9, 6726 Szeged, Hungary; akos.mesterhazy@gabonakutato.hu

**Keywords:** fusarium head blight, resistance components, common resistance, multitoxin contamination, food safety, FHB variety registration, FHB resistance testing methods, Fusarium-damaged kernels (FDK), resistance to toxin accumulation

## Abstract

The term “Fusarium Head Blight” (FHB) resistance supposedly covers common resistances to different *Fusarium* spp. without any generally accepted evidence. For food safety, all should be considered with their toxins, except for deoxynivalenol (DON). Disease index (DI), scabby kernels (FDK), and DON steadily result from FHB, and even the genetic regulation of *Fusarium* spp. may differ; therefore, multitoxin contamination is common. The resistance types of FHB form a rather complex syndrome that has been the subject of debate for decades. It seems that resistance types are not independent variables but rather a series of components that follow disease and epidemic development; their genetic regulation may differ. Spraying inoculation (Type 1 resistance) includes the phase where spores land on palea and lemma and spread to the ovarium and also includes the spread-inhibiting resistance factor; therefore, it provides the overall resistance that is needed. A significant part of Type 1-resistant QTLs could, therefore, be Type 2, requiring the retesting of the QTLs; this is, at least, the case for the most effective ones. The updated resistance components are as follows: Component 1 is overall resistance, as discussed above; Component 2 includes spreading from the ovarium through the head, which is a part of Component 1; Component 3 includes factors from grain development to ripening (FDK); Component 4 includes factors influencing DON contamination, decrease, overproduction, and relative toxin resistance; and for Component 5, the tolerance has a low significance without new results. Independent QTLs with different functions can be identified for one or more traits. Resistance to different *Fusarium* spp. seems to be connected; it is species non-specific, but further research is necessary. Their toxin relations are unknown. DI, FDK, and DON should be checked as they serve as the basic data for the risk analysis of cultivars. A better understanding of the multitoxin risk is needed regarding resistance to the main *Fusarium* spp.; therefore, an updated testing methodology is suggested. This will provide more precise data for research, genetics, and variety registration. In winter and spring wheat, the existing resistance level is very high, close to Sumai 3, and provides much greater food safety combined with sophisticated fungicide preventive control and other practices in commercial production.

## 1. Introduction

FHB is one of the most destructive diseases affecting wheat. Heavy epidemics can cause very high yield losses in individual fields, and large regions can be devastated where not only is the yield reduced, but also the remaining yield suffers severe quality losses due to toxins [1,2,3]. Consequently, the toxin contamination may cause severe financial damage. Resistance to FHB is a complex phenomenon. “Fusarium head blight”, caused by different *Fusarium* species, is a very commonly used term, with nearly all titles of articles in this field containing these three words. As many *Fusarium* species are involved, it is essential to understand whether FHB resistance has a common resistance to all *Fusarium* species, or whether different genes regulate resistance to different species. I suspect that common resistance is the consensus in the research [4,5], but only a few papers support this [6,7,8]. The resistance mechanisms can be divided into active and passive (morphology based) forms [7]. Passive forms include plant height and longer and looser heads that dry out rapidly, and so inhibit disease development. The long internode between the head and flag leaf is also important because the splashing raindrops from the flag leaves reach less than the heads standing far above them. The active forms are those where physiological processes and genetic interactions determine the response of the plants. They are **not** included in the described resistance types, but all genes belonging to this group regulate physiological processes and support an effective protection against the pathogens and their toxins. Their combined application is part of the breeding work [7]. 

We need greater care in our use of terms. It is generally accepted that resistance to disease and toxin accumulation are synonymous, and that this would justify the negation of toxin analyses and basing selection on visual symptoms. The significance of FDK was recognized by Schroeder and Christensen [9], but the data were controversial, so the FDK trait was forgotten. This is reasonable insofar as resistance is connected to the primary appearance of the pathogen or its characteristic symptoms, but in the toxigenic fungi, the resistance to toxin accumulation is the most important trait and is not necessarily connected to the resistance level. This is why most papers only deal with visual symptoms and why the extent to which this resistance influences toxin accumulation, with exceptions, has not been documented. When a person sells wheat grain, the only question is (in addition to baking or other quality traits like protein), how high the toxin content is. This means that breeding should concentrate on reducing toxin contamination rather than only paying attention to the visual symptoms of the ear. As research has established that the QTL regulation of the three traits can be different [10], a higher resistance (fewer symptoms) does not automatically guarantee low toxin contamination. 

Types 1 and 2 are the original designations used in this paper, therefore all other designations, like type I and Type II, are false, so their use is not suggested (described by Schroeder and Christensen) [9]. Type 3 (resistance of kernels to Fusarium infection), Type 4 (resistance to DON accumulation), and Type 5 (tolerance [7,8,11,12]) are described later. We can add anther extrusion with solid proof [13,14,15], but we do not know whether this extrusion is related to the resistance components known to date or if it can be classified as a component of resistance Type 1. Resistance Type 1 is called FHB resistance, Type 2 resistance is also FHB resistance, as is resistance to DON accumulation, with three different meanings for the same term. Are they really different or not? In past decades, a huge amount of the literature was published [7,8,12,16,17,18,19,20,21,22,23,24]. However, this question has not been fully addressed. In this paper, I consider the development of more resistant cultivars with less mycotoxin contamination as the ultimate objective. It is also very important to better understand their complicated relations and functions. We are also faced with the problem of emerging mycotoxins and the food safety complications of the many *Fusarium* spp. presenting a considerable challenge. 

Considering global warming the dryer and hotter seasons do not support FHB epidemics in Hungary, but even under these conditions one or two national and several regional and more local ones will occur [25] in every ten years. In regions like Northern Europe, the rate of *F. graminearum sensu stricto* will increase, and the same is true in the southern areas of Siberia and North China. The areas where the disease is year around present like in Middle China and others, the present conditions remain, therefore, heavily rainy, and warm, in other jeopardized regions hit now by frequent epidemics, the higher resistance remains a crucial precondition of successful production [26]. Most of the toxin forecasts do not consider the role of resistance to disease and toxin accumulation in balancing the negative effects of climate change; therefore, the forecast scenarios seem to me worse than they could be [26,27,28]. We found in South Hungary, aflatoxin in wheat (Berenyi et al. unpublished), and the forecast of Battilani et al. [28] for wheat also seems to be real. The forecast for the whole Carpathian Basin for maize severe aflatoxin contamination increase for maize shows that and South Germany and part of Austria will suffer under a 2.5 °C increase. When a 5 °C increase occurs, even Helsinki will be jeopardized. Aflatoxin is in Hungary now on maize, and wheat is also under threat. As resistance data from varieties and hybrids are not available, they cannot be considered in the forecasts. However, the disease resistance and resistance to toxin contamination should be treated more seriously. An analysis of the 1993–2001 FHB epidemic in the US proved that the more resistant cultivars reached only 50% of the DON contamination found in susceptible varieties [29].

In this review, the whole mass of the mycotoxin problem of wheat is considered from various points of view, concentrating more on the food safety effects caused by other *Fusarium* spp. and their toxins beyond the *F. graminearum* and the problem of DON, which is also a mostly unsolved problem. 

## 2. Losses Due to FHB in Wheat Production

The total global wheat yield in 2018 was 774 million metric tons (MMT). As preharvest losses are estimated to be about one-third of the harvested yield, the global yield capacity is 1161 MMT [30]. Of this 441 MTT is used as healthy grain (38%), the rest is absent or has restricted quality and a poor market position, if any (Table 1). The total pre- and postharvest loss is 720 MMT. The mycotoxin contamination was estimated by Mannon [31] as 25% of global cereal production; the FAO estimate is the same as that was confirmed by Eskola et al. [32]. Mesterhazy et al. [30] estimated the loss at harvest as 10% above the limit value. By adding the 6% preharvest yield loss estimate and one-third (6.7% estimate) of storage-based estimates [33,34] of the storage loss (20%), 22% total damage is probable by discounting the period before harvest and severe damage by toxins is in the periods before and after harvest, but research is more detailed in terms of its origin; a better diagnosis may help different approaches to decrease losses. The numbers can be controlled. It is hoped that after reforming the cereal production system, we could obtain detailed and highly reliable data. Applying the data on wheat, we face a 173–179 million metric ton loss depending on whether one compares the total yield with preharvest and postharvest together, or the total loss is counted for the whole production chain. From this loss, a 50% reduction of losses in the long term is possible, which is enough to feed about 1 billion more people. When the other losses are also reduced, the amount saved grain can be significantly higher. We should say, this production system is not sustainable, but the losses can be significantly reduced. 

## 3. The Causing Agents

The *Fusarium* species produce a large number of mycotoxins, differing widely in nature, many of which are responsible for the production of even 5–10 different toxins, including masked forms [35]. There is another problem, one that can be seen in Table 2, which lists the most important *Fusarium* spp. from wheat in Europe [36]. One species can produce many mycotoxins and different species may produce the same mycotoxins. Therefore, from the toxin data, it is unclear what is the causing agent, and even when we know the causing agent, the actual toxin contamination cannot be predicted. The number of toxins is now significantly higher; a number of masked and other toxins were identified with very high toxicity (DON, NIV, T-2, HT2, ZEN and among their masked variants 3-acety-DON should be mentioned) [36,37]. *F. crookwellense* Burgess, Nelson and Tousson (syn. *F. cerealis* (Cooke) Sacc.) is a dangerous NIV producer. 

In Northern Europe [38] the leading *Fusarium* species belong to the *F. avenaceum/arthrosporioides/tricinctum, F. graminearum/culmorum/cerealis* and *F. poae/sporotrichioides/langsethiae* species groups. Based on morphology. *F. langsethiae* Torp and Nirenberg are dangerous T-2 and HT-2 producers together with *F. sporotrichioides*. In 2010 *F. graminearum* replaced *F. culmorum*. Uhlig et al. [39] noted that *F. avenaceum* is not an important BEA, while *F. tricinctum* produces enniatins. O’Donnel et al. [40] revolutionized the *Fusarium* taxonomy. The *F. graminearum* clade is settled, *F. graminearum* sensu stricto is dominant in Europe and North America, but several species like *F. asiaticum* are known as NIV producers. Nivalenol producers were rarely found in *F. graminearum* and *F. culmorum*. The *F. avenaceum* is also developed to a *species complex* [41], and these species are enniatin producers, BEA was produced only by one isolate, so its significance does not seem high. AOD: 2-amino-14,16-dimethyloctadecan-3-ol is produced at a rather high level, though only seven isolates produced traces that could not be quantified. These new findings make the toxin reduction even more complicated. 

The question is how far this problem can be treated by breeding. There is another important aspect to Table 1. In North and Central Europe five *Fusarium* species were considered as highly pathogenic causing significant epidemics and toxin contamination (*F. graminearum*, *F. avenaceum*, *F. culmorum*, *F. poae* and *F. equiseti* should be mentioned [36]) and based on [38] *F. langsethiae* with its T-2 and HT-2 toxins. In the south *F. graminearum* is the most frequent, and the significance of the others is moderate or not significant [36]. Wang et al. [42] summarized toxin occurrence data in Europe for the period 2010–2020 and predicted an increase in deoxynivalenol, zearalenone, T-2 toxin and HT-2 toxin, fumonisins, aflatoxins, and/or ochratoxin A contamination. Resistance data were not included in the calculation. This could be explained by a lack of data, which is highly probable. However, in this case, a discussion of the problem is well supported. The situation is not stable: between epidemic and non-epidemic years, there might be significant differences in disease severity and species composition [43]. 

For this reason, a broader resistance working against more *Fusarium* spp. would be necessary in both Southern and in the Northern Europe. However, the resistance is advantageous in epidemic years; therefore, to ensure food safety, a broad spectrum of resistance is necessary to control infections by different *Fusarium* species. The fact that the resistance of Sumai 3 is good in all the regions it was tested supports a working hypothesis that the resistance is maintained even when the species composition is significantly different in the wheat production regions. It is very probable that the warmer seasons in the north may decrease the number of ruling species in Mid and Northern Europe. In several parts of China *F. asiaticum* is dominant (it is also an NIV producer), followed by *F. graminearum sensu stricto*, whereas the others occur only rarely [44], and the most infected areas are Hebei, Hubei, Jiangsu, and Sichuan. 

**Table 2 toxins-16-00031-t002:** Mycotoxigenic Fusarium species isolated from wheat grains in Europe [36].

Species	Geographical Incidence		Mycotoxin
	North/Center	South	
*F. graminearum* Schwabe	+++	+++	DON, NIV, ZEN, AcDON, FUS
*F. avenaceum* (Fr.) Sacc.	+++	++	MON, BEA, ENS
*F. culmorum* (W.G. Smith) Sacc.	+++	++	DON, ZEN, ZOH, NIV
*F. poae* (Peck) Wollenw.	+++	+	NIV, BEA, DAS, FUS, ENS
*F. equiseti* (Corda) Sacc.	+++	+	DAS, ZEN, ZOH
*F. tricinctum* (Corda.) Sacc.	+	+	MON
*F. cerealis* (Cooke) Sacc. (*F. crookwellense*, Burgess, Nelson and Tous.)	+	±	NIV, FUS, ZEN, ZOH
*F. sporotrichioides* Scherb.	+	±	T2, HT2, T2ol, NEO
*F. acuminatum* Ellis and Everhart	±	±	T2, NEO
*F. subglutinans* Wollenw. and Reinking	±	-	MON
*F. solani* (Mart.) Sacc.	±	-	-
*F. oxysporum* Schlecht.	±	-	-

AcDON = Monoacetyl-deoxynivalenols (3-AcDON, 15-AcDON); BEA = Beauvericin; DAS = Diacetoxyscirpenol; DON = Deoxynivalenol (Vomitoxin); ENS = Enniatins; FUS = Fusarenone-X (4-Acetyl-NIV); HT2 = HT-2 toxin; MON = Moniliformin; NEO = Neosolaniol; NIV = Nivalenol; T2 = T-2 toxin; T2ol = T-2 tetraol; ZEN = Zearalenone; ZOH = Zearalenols (α and β isomers). The descriptors of the species were given according to Booth [45] and Leslie and Summerell [46], as the authors of this table used mostly these taxonomies or their earlier versions. In several cases, Refai et al. [47] was also consulted.

The *Fusarium* species causing FHB are numerous in countries where data coming from a high number of species were identified [43,48,49]. We were not interested in collecting data for all *Fusarium* spp. that do not cause significant damage [43]. In this survey, there was a low rate of *F. poae* compared to the high rate found in Hungary by Xu et al. [50]. In this study, PCR markers to identify *Fusarium* spp. and their rates were used. It seems that this aspect should be considered when we discuss the significance of a *Fusarium* species. For this reason, only comparable data (only isolates from clearly scabby grains or only a sample representing the whole grain lot) should be analyzed together. In the studies, no toxin analyses were made, so the food safety aspects remained unknown. Nonetheless, there is little doubt that *F. graminearum sensu stricto* is the dominating species of the FHB epidemics in most wheat-growing regions. 

The multiple nature of the pathogens is supported by the data summarized by Leslie et al. [51]. According to the latest review [52], in Europe, *F. graminearum*, *F. culmorum*, *F. avenaceum*, *F. poae*, *F. tricinctum*, and *Microdochium majus* (syn. *Fusarium nivale*) are the dominant species, along with many others. In Canada, *F. avenaceum*, *F. equiseti*, *F. graminearum*, *F. poae* and *F. sporotrichioides* are mentioned as major species. It is understandable that breeding efforts have been made against these ruling or dominant species, implicitly supposing some art of common resistance against them [53]. 

Many papers have been published on the member species of the *F. graminearum clade* [54,55,56] their distribution is very variable globally, and in some countries, even 4–5 species may occur. Dong et al. [57] report a massive presence of *F. asiaticum* in Chinese wheat–rice regions where rice and wheat follow each other, so the wheat is exposed to this species that produces mainly nivalenol. At present, 16 monophyletic species have been described in *F. graminearum*, forming the *F. graminearum* clade [54,58,59], based on several molecular traits, sometimes with differing toxin profiles. It is unclear whether resistance to them is connected to *F. graminearum sensu stricto* or to each other. Similarly, the biological identity of these new monophyletic species remained unknown [51].

The question is whether resistance to these species is connected or not. In the latter case, we should breed for resistance against all separately and later pool them. The problem is challenging and needs clarification. We had the same problem after detecting the many members of the *F. graminearum* clade. In several regions of the world like South Africa, South America, China, etc., more members occur at the same time, but it is not clear whether resistance to *F. graminearum sensu stricto* and *F. asiaticum* has the same resistance background or a different one [54,55,56]. At present, in Europe, *F. graminearum sensu stricto* is actually the ruling species [56]. As ruling species may change in time, for example, in North Europe the *F. culmorum, F. avenaceum* or *F. langsethiae* Torp and Nirenberg will be replaced by *F. graminearum*, resistance relations become important [36,60,61]. Senatore et al. [62] tested 70 Italian wheat samples for multitoxin analyses: 24 different species were identified, and the most frequent were *F. graminearum* and *F. avenaceum* with about equal weight followed *F. poae*, *F. proliferatum* by PDA + tef1a, and 24 *Fusarium* spp. were detected by the DFB + mc method. The presence of the multispecies infection is a common, natural phenomenon. Of them, eight were selected for qPCR tests. Of them four dominated the picture, *F. graminearum*, *F. avenaceum*, *F. culmorum and F. langsethiae*. The closest correlation between fungal mass and toxin was found for DON (r = 0.7612), *F. avenaceum*, and moniliformin (r = 0.676), with enniatins r = 0.484), *F. langsethiae*–HT-2 toxin (r = 0.784) *F. tricinctum* and moniliformin (r = 0.548). For *F. culmorum*, *F. sporotrichioides* and *F. poae* only low and mostly non-significant correlations were found. It seems that for the species giving good correlations with toxin contamination, the qPCR can be important for research purposes, for the rest, other solutions should be looked for like direct multitoxin analysis. For us, the multi-Fusarium infection is a fact, the multi toxin occurrence is also a fact, and, in several cases, the qPCR can be useful to forecast toxin contamination. As the species occurrence rate also depends on the year and the season, a two-part question arises: will our variety have more or less resistance to *F. graminearum*, and will it maintain its resistance when other species dominate the epidemic? There are only sporadic data on this problem. The answer to this question largely determines the outcome of breeding work.

In conclusion, we encounter varying numbers of important species in the northern, central, and southern regions of Europe. The species composition of the Fusarium population is steadily changing, from season to season and from location to location. The ruling species is mostly *F. graminearum sensu stricto*; therefore, nearly all breeding programs concentrate on this. Is this enough? Based on Table 2, another conclusion is possible. In the central and northern regions, more *Fusarium* species cause more severe diseases than in the south, possibly because the conditions there are less suitable for disease and toxin development under warm and mostly dry conditions. In continental-size countries, like the USA, Canada, Brazil, India and China, high variability is also found, and the species composition can be very diverse. For this reason, we should have information about the situation in the case of multiple infections. When susceptibility is high, multiple infections and toxin contamination can be possible; therefore, a multitoxin analysis will be necessary to estimate the mycotoxin exposition correctly. Is this somehow connected to resistance? For a discussion of this, see Section 5.

## 4. Multi Mycotoxin Contamination in Freshly Harvested and Stored Wheat

It seems that we do not simply encounter a single mycotoxin occurrence in our wheat and wheat products; usually, there are more [62]. Varga et al. [63] reported from Hungary between 1994 and 2002 9689 toxin analyses from cereals, including wheat. From the Fusarium toxins DON, zearalenone and T-2 toxins were screened. For wheat, 2300 grain samples were tested between 1997–2002. The national means were between 93 and 903 μg/kg for DON, between 3.5 and 18 μg/kg for ZEN, and between 11 and 36 μg/kg for T-2. The highest data were recorded from 1997 to 1998, which were epidemic years. Even the occurrence on *F. sporotrichioides* was low, and T-2 occurred consequently, in several cases, above the limit. Zearalenone was a surprise, in freshly harvested wheat we could not identify ZEN earlier, even at a very high DON contamination. They [63] compared the mycotoxin contamination of well and badly stored maize. The regular samples did not show higher mean values than the EU limit for FUM, DON, T-2 and ZEN in any year. The general mean values were, however, 4.44-fold, 8.01-fold, 2.16-fold and 5.21-fold higher in terms of date in badly stored maize than in the well stored lots. The case seems to be similar in wheat [64] and the a_w_ was the most important agent to increase DON and ZEN production. An a_w_ of 0.95 yielded in 10 days 5 mg/kg; an a_w_ of 0.97 (water activity), in the silo top yielded a 10 mg/kg increase on the sixth day. Portell et al. [64] noted that, despite the importance of the topic, the literature is sparse. Magan et al. [33] and Mylona et al. [34] stressed the importance of suboptimal storage conditions for the significant increase in mycotoxins in stored wheat, not only for so-called storage fungi but also because field-originated fungi can produce significant amounts of different mycotoxins under a microbe friendly environment. Magan et al. [33] also mentioned this aspect in relation to experimental data, concluding that **successful storage needs healthy grains to upload the silos**. Nesic et al. [65], in their review, cited many papers reporting about multitoxin contamination in cereals and wheat and this could be generally valid for all cereal crops. Further, 60–80% of the samples contained mycotoxins, and 75% of them contained two or more in a sample. The tests were made on preharvest and postharvest samples. For us, the preharvest contamination is strong and determining because for the plant breeding this information is decisive. From the postharvest contaminated samples, the preharvest origin cannot be proven, only supposed. Sadhasivam et al. [66] tested 10 mycotoxins from 34 stored wheat samples: FB1 was found in 9, FB2 in 8, DON in 8, and ZEN in 5 samples. These last three mycotoxins are normally produced in fields, but their field or storage origin could not be deduced from the articles.

A sample that does not contain DON does not mean that other mycotoxins could not be present above the limit. As knowledge is growing, the conviction is stronger to have practices that can manage this situation. The key to this is the presence of highly effective, rapid, and cheap multitoxin methods that allow for the control of the trucks coming from the field to separate the excellent quality from bad. These methods can help breed for resistance and evaluate the effect of different practices on general toxin reduction.

The problem is that we do not have cloned genes and their products to perform this work. Another problem is the high number of possible toxins. As the paper mentions 11 different toxins of a possible 500 (approx.), the problem is to understand the nature of disease resistance and resistance to toxin accumulation to decide what the nature of the task is. Field fungi are supposed to stop their activity in storage and toxin synthesis, and in well-operated bins or other facilities, this is the case [67,68]. For practical reasons, the limits should be considered to make a distinction between an acceptable and non-acceptable amount of toxin with one exception, that is, baby food quality, which should be ensured for children and young animals. For DON, this limit is 0.2 mg/kg; to consider this, the rate of contaminated lots is possibly not 10 or 25% but could be 30–40%, or higher. The problem is that the agriculture cannot secure this low toxin level in epidemic years, and sometimes not even in regular years when susceptibility is generally widespread. This explains the compromise between physiological needs and tolerable toxin intake. The real problem is that despite the huge investment in reducing toxin contamination, every epidemic teaches us that the success is rather moderate. The conclusion is that only healthy grain should be stored under optimal and controlled storage conditions online, but in epidemic years, higher or high resistance is a precondition of low- or no-loss storage outlook.

Thus, resistance breeding has responsibility for the storage of healthy grains.

## 5. Multitoxin Presence, an Emerging Problem

LC-MS/MS analytics is a rather new technology. By developing different multitoxin methods, it has become possible to identify even a large number of mycotoxins and compounds with suspected toxigenic abilities from the same sample [35,37,69]. The question raised is how much relevance this has for a possible plant-breeding problem. The introduction of the recent analysis of mycotoxin made it possible to observe hundreds of compounds in a sample [70]. In 138 samples, 46 mycotoxins were identified, indicating a very wide range of potential contamination [71]. A recent review of HRMS data that covered urine, blood, and food samples [72] concluded that non-targeted analyses are better for research, because they can detect all mycotoxins, not only a handful of the most important ones. In a study in Sweden [73], 1105 human blood samples were screened for (DON, 4.8%), DON-15-β-D-O-glucuronide (DON-15GlcA, 9.1%), dihydro-citrinone (DH-CIT, 0.5%), HT-2-glucuronide (HT-2-3-GlcA, 0.1%). DON was found in the blood of people who consumed oats, dietary fiber and whole grain, and enniatinB (EnB) was detected in all samples. Izzo et al. [74] conducted targeted (six toxins) and non-targeted analyses of the same 56 milk samples. In addition to the 5 mycotoxins, 49 pesticides and 11 veterinary drugs were identified. However, they recommended targeted identification for routine analyses. In Italian barley samples, Beccari et al. [75] identified from 43 malting barley samples HT-2 toxin (28%), followed by enniatin B (24%), enniatin B1 (22%), T-2 toxin (17%) and nivalenol (15%). The real composition of humans and animals can be identified by multi-mycotoxin tests from blood or urine. Abia et al. [76] identified (from 175 urine samples) and in 110 (65%) contained aflatoxin M1, fumonisin B1, ochratoxin A, and total deoxynivalenol alone or in combination. Foerster et al. [77] reported results from 172 participants of the Maule Cohort and found DON at 63%, AFB1 8%, α-zearalenol 41.%, β-zearalenol 3.5% and ZEN 0.6%. De Santis et al. [78] screened 5753 publication and selected 176 papers for evaluation for mycotoxins as biomarker data in human populations. Of the Fusarium toxins, more than four countries had DON, FBs, ZEN, and OTA, and the most mycotoxins were found in Germany (7), Estonia (6), Italy (5), and Sweden (5). One mycotoxin per country was detected in Austria, Croatia, Norway, and Great Britain. In addition to DON, ZEN, FBs, T2/HT2 and NIV were critical in a significant part of the samples. Schmidt et al. [79] detected positive cases of aflatoxin M_1_ (AFM_1_ 18%), alternariol monomethyl ether (AME 2%), alternariol, citrinin (CIT 14%) and its metabolite dihydrocitrinone (DH-CIT 18%), fumonisin B_1_ (FB_1_ 100%), ochratoxin A (OTA 38%), and zearalenone (ZEN 10%) from 50 human urine samples. Viegas et al. [80] found for 25 workers deoxynivalenol-glucuronic acid conjugate (60%), aflatoxin M1 (16%), enniatin B (4%), citrinin (8%), dihydrocitrinone (12%) and ochratoxin A (80%).

This suggests that the number of mycotoxins and their concentration is much greater than previously supposed. Valenti et al. [81] reported several papers with the coexistence of DON and enniatins including synergistic, antagonistic, and additive effects. Nicholson et al. [82] reported enniatin synthetase presence with primers Ensyn6065F/Ensyn7229R in nine *Fusarium* spp., including *F. avenaceum*, *F*, *chlamydosporum*, *F. sambucinum*, and *F. tricinctum*, but not in *F. graminearum*, *F. culmorum*, *F. equiseti*, *F. poae* or *F. proliferatum*. It is important that *F. tricinctum* is an enniatin producer [39]. It seems that the enniatins have many producers, which also makes this problem harder.

Multi-mycotoxin analyses in durum pasta samples were conducted screening 27 samples of durum wheat pasta for 17 mycotoxins [83], 9 of which occurred in the samples with more than 80% incidence, mainly DON (100%), 3 + 15 AcDON (86%), HT-2 toxin (90%), zearalenone (93%), enniatin A (93%), and enniatin B (90%). The high occurrence rate of the enniatins poses a new problem: how to control them. Few of them feature in the plant pathology literature, but the data should be considered in discussions of food safety. The question is how these findings are helpful for breeding in terms of detecting resistance to the causative agents. We should recognize that OTA is a postharvest mycotoxin; its preharvest occurrence is unclear. A clarification would be needed. In addition to mycotoxins, their masked variants can also create serious food safety problems [37,84]. We know now that breeding for low DON results also lowers masked toxins for DON; therefore, a specific breeding program for the masked forms does not seem to be necessary [85] We should remark that the kinetics of toxin accumulation depends on variety, isolate of *F. culmorum*, year for NIV, and DON in the 0–28 days after inoculation. This underlines our experience that, at present, genetically valid data in a significantly larger data are more necessary than generally supposed.

As many *Fusarium* spp. infect wheat with mostly differing toxin synthesis, the occurrence of many mycotoxins is unsurprising and even the same lot can have different mycotoxins being detected at the same time. Their occurrence in blood and urine supports the multi-Fusarium and multitoxin occurrence. On the other hand, most papers stress in addition to the often high yield loss the significance of DON reduction in resistance screening, however, about 10–15% of the papers only analyze visual symptoms instrumentally [7,86,87,88], which can make the remaining yield to worthless waste. When one percentage of visual head severity (DI) or FDK symptoms can show a 10-fold or higher differences in DON contamination, conclusions for the resistance to DON accumulation are problematic [7,87,89,90]. The questions are raised: what is the value of *F. graminearum* and DON resistance to toxin accumulation in wheat when multitoxin contamination is normal, and are the occurrence of non-*F. graminearum*-originated mycotoxins also normal? In conclusion, the food safety aspect of the head blight caused by many *Fusarium* spp. is much wider than the DON problem alone is. It should also be clarified, how far plant breeding can contribute to the multitoxin reduction, and which additional practices can help to keep mycotoxin contamination below the limit.

For these reasons, in this review, we also investigate the resistance of wheat to the whole FHB complex also including the *F. graminearum* clade.

## 6. Resistance to *Fusarium* spp.

Most papers investigate one *Fusarium* species, and in the vast majority of cases, it is *F. graminearum.* van Eeuwijk et al. [91], in a multilateral FHB test working with isolates of *F. graminearum* (6), *F. culmorum* (9), and *F. nivale (M. nivale*) (1) on 25 European winter wheat genotypes from Austria, France, Germany, Hungary, and the Netherlands, found no specificity within the European *F. graminearum* and *F. culmorum* population, and resistance to the two *Fusarium* species seems to have the same background. Therefore, it is not necessary to select shuttle breeding for FHB resistance to these two species. It seems that the resistance background to *F. nivale* (now *Microdochium nivale*) is the same as that of *F. graminearum* or *F. culmorum*. It is remarkable that common resistance can exist in fungal pathogens that do not belong to the *Fusarium* genus. As we had only one *F. nivale* isolate, this should be checked and confirmed. One of the conclusions was that the close correlations between responses of the genotypes to the different isolates of the pathogens would support the approach of using only one isolate in resistance testing. As DON data were not screened, the food safety relations could not be judged. Subsequent tests confirmed the non-specificity of resistance to *F. graminearum* and *F. culmorum.* They correlate very closely; indeed, the resistance to them is probably the same [16,21,92,93]. This applies not only to the symptoms but also to the resistance to toxin contamination. Mesterhazy et al. [94,95] achieved very similar results when testing cultivars and lines. Other tests revealed very distinct resistance differences between these two species supplemented with *F. sambucinum*, *F. sporotrichioides,* and *F. verticillioides* (1998, 1999). In this case, not only head symptoms but also FDK and DON were monitored (for non-DON producers, the DON was the natural background contamination at a low level [94]). In this test we counted the DON production for 1% FDK infection (Figure 1). As this is between 0.2 and 3.7 mg/kg, about an 18 fold difference this means that the same infection severity can mean highly different DON values for a percent of FDK. This lessens the correlation between visual symptoms and DON contamination, and it is an argument to measure DON. Even the correlations are medium or higher as we have to point out problems in forecasting behavior of individual varieties. We first described this phenomenon, but nothing is known about its genetic background [94]. 

Another test from this paper used summarized results (2001–2002) for 2–2 isolates of *F. graminearum*, *F. culmorum*, *F. avenaceum* and *F. sporotrichioides*, and one *F. sambucinum* isolate was also added (the other *F. sambucinum* isolate did not infect). The data indicated common resistance to the different *Fusarium* spp. (Table 3). In a test of 26 wheat varieties and lines, the isolate mean values showed that *F. culmorum* 12551 with an average FDK of63% (8.58–98.17%) was the most pathogenic, while the least pathogenic was *F. sporotrichioides* with an average FDK of 11.33% (0.75–47%). The most resistant was Sumai 3 with an FDK mean of 2.5% (0.08–8.58%), and the most susceptible was the line SJ981153 with an FDK of 66.7% (26.75–98.17%). The data clearly show that all correlations between the isolates or inocula were practically the same as the interspecies correlations. In *F. graminearum* and *F. culmorum*, the aggressiveness of the isolates was very similar; in *F. avenaceum* and *F. sporotrichioides*, the aggressiveness was significantly lower. The differences between isolates were larger in several varieties. As only the measuring of DON was possible, the specific toxin responses of *F. avenaceum* (moniliformin), *F. sporotrichioides* (T-2 Toxin), and *F. sambucinum* the diacetoxyscirpenol (DAS) [96] were not tested. Therefore, only resistance behavior could be compared for the different *Fusarium* spp. beyond the DON-producing *Fusarium* species.

Table 2 also provides another conclusion. As the resistance is well connected to different *Fusarium* spp., it means that the most susceptible genotypes to *F. culmorum* have very severe infections (98%) also against the less pathogenic *F. graminearum* 87%, *F. sambucinum* 90%, *F. avenaceum* 46% and *F. sporotrichioides* 48%. This means that the food safety problem in Fg/Fc-susceptible wheat varieties makes the otherwise less pathogenic *Fusarium* species a dangerous food and feed risk agent. The correlations between variety responses to the three most aggressive *Fusarium* species are above r = 0.90, their relationship with the *F. avenaceum* and *F. sporotrichioides* is looser but also highly significant. This may be due to the reduced aggressiveness level, as at lower aggressiveness, the genotype differences are less reliable. In several cases, genotype-specific effects are present like in Zugoly or Öthalom with 1.67% and 5.25% for *F. avenaceum*, respectively, but their origin is unknown; it could be ecologic, genetic or a combination of both. As we measured only DON, we do not understand the toxin production and regulation of non-DON producers. We found that in most genotypes, FDK and DON contamination correlate, but a number of exceptions can be identified [92,95]. As the optimum temperature is different for toxin synthesis, the T-2 and HT-2 optimum temperature for toxin production of *F. sporotrichioides* is 12.1 °C [97]; for zearalenone, it is 20 °C [98] and for *F. avenaceum*, it is between 5 and 15 °C [99]. They can cause more severe infection, whereas *F. graminearum* at these temperatures causes only sporadic infection. In such cases the low or no-DON contamination does not automatically mean a similar toxin contamination for these fungi.

The variance for the genotypes across the five *Fusarium* spp. shows high differences. The lowest variance was shown for Sumai 3 (12.3) and Nobeoka Bozu (18.7). Less than σ = 100 wheat genotypes were measured for Wuhan 4B, and 3 wheat genotypes were measured for the Szeged program. Frontana had 170. At higher susceptibility, the variance increases, with the highest value found for Zugoly (1591).

As mentioned above, we have no clear proof for the regulation of disease resistance and resistance to toxin accumulation for the three less pathogenic species. The verified high number of mycotoxins exacerbate this problem. From a food safety perspective, it seems that the high susceptibility to *F. graminearum* significantly increases the contamination of other mycotoxin producers, e.g., it might significantly increase food and feed safety risks. The presence of the different toxins and their interactions exacerbate the problem further. This study should be carried out as soon as possible to create a solid base for the work.

We mentioned the discovery of the *F. graminearum* clade, and its many new species were described. As they can be present in the same regions at a significant rate [44,54,58,59], the resistance to them is a vital concern not only for theoretical research but also for practical breeding. Eight members of the *F. graminearum* clade supplemented with *F. culmorum* isolates were tested, based on visual symptoms, for FDK and DON contamination [100]. The correlations between species are, in all aspects, close and significant at *p* = 0.001, with the only exception being *F. boothii*. The representative isolates were provided, one for each species by O’Donnel in 2002. We have tested them on 12 genotypes from very high resistance (Nobeoka Bozu) to highly susceptible (Kalász). The *F. meridionale* and *F. mesoamericanum* had much weaker correlations than the other more aggressive species with the same results, and the low pathogenic species differentiated much less than other varieties as shown in Figure 1 of the original publication. However, the resistance to the different species correlated well. We do not think that this is not a genetic deviation but rather the consequence of the lower aggressiveness. This agrees well with earlier results, and it seems that there is a general resistance to different *Fusarium* spp. When this is generally true, it means that the susceptible and highly susceptible varieties, even the less pathogenic Fusarium species, can experience higher disease severity and higher toxin contamination. The most pathogenic isolate belongs to *F. culmorum*, but two other isolates, according to the FDK data, are close to *F. graminearum*, and we found similar pathogenicity and FDK for *F. acacia-mearnsii*, *F. asiaticum* and *F. cortaderiare*. However, other than their DON production, they are significant nivalenol producers; based on DON production, they do not seem risky. The situation is very different when nivalenol is calculated with its ten-fold greater toxicity than DON; therefore, it can cause a high food safety risk. This toxin should also be checked in NIV regions as mostly occur together. For this reason, we should sum the amounts of the different trichothecenes and express their toxicity in DON equivalence. When, for example, we have 10 mg/kg DON and 1 mg/kg DON, the NIV data should be multiplied by 10 and summed so that the DON equivalence will be 20 mg/kg. When we have more trichothecenes with masked versions, their toxicity can also be expressed in a similar way. In the future, it will be better to express nivalenol in a DON equivalent to make it more comparable, and the total trichothecenes should be provided in a DON-equivalent manner. This is important to assess the real mycotoxin risk of a given grain lot. Of course, the other non-trichothecene mycotoxins should also be considered. In addition, animal tests should also be conducted to see how this works in response to the different animal species and their human relations are also important research considerations.

In conclusion, the many new species within the *F. graminearum* clade will not create additional breeding problems. However, their NIV producer members may provide new food safety problems with significant NIV production. We should also consider that the different *Fusarium* spp. have different ecological needs, for example, the optimum temperature is lower than that of *F. graminearum*; therefore, these species could cause significant toxin contamination without significant DON contamination. We should remark that, at present, this paper is the only one in this context; therefore, future research work should be conducted to obtain more data and knowledge on the topic.

The discovery of the masked DON variants [37] made the food safety aspects of DON more complicated. DON-3-glucoside is a nontoxic variant of DON, and one DON-glucosyltransferase is located in the *Qfhs.ndsu-3BS* region. DON is considered to be a virulence factor for *F. graminearum*. It was concluded that the detoxification of DON may be a useful form to reduce DON pressure in the grain and increase food safety. This is problematic for many mycotoxins [84]. Lemmens et al. [101] compared the DON resistance of plants from the CM-82036/Remus mapping population and found that DON contamination was closely correlated after artificial inoculation with the DON resistance of plants treated with DON toxin. It was later found that this process can be reversed, so the DON-3-glucosyde can be split again into DON and glucoside, creating additional food safety problems [37]. The next question concerned how the masked mycotoxins influence breeding. In their review, Lemmens et al. [101] found that DON contamination of the grains in the resistant plants is much smaller than that in the susceptible ones, and, therefore, the masked DON forms in the resistant plants do not pose a serious risk, even though their rate is higher than that in the susceptible ones. All tested cultivars were able to convert DON to DON-3-glucoside, not only that located in 3BS QTL. Its rate is sometimes substantial, and may reach 35%, but it depends on both genetic and environmental factors. DON-3-glucosyde content is highly correlated with the symptom severity of heads treated with DON, and FHB resistance reduces both DON and DON-3-glucosyde in the grain, no additional breeding effort being necessary. When the highly resistant plant is ready, the masked DON can be checked to avoid food safety problems from the decomposing DON-3-glycoside. However, this is only one example, and its validity should be checked for the other QTLs that play a role in this process.

The CM-82036/Remus population was checked in an FP6 EU Project where 24–24 genotypes were selected by Buerstmayr and Ruckenbauer (IFA Tulln Austria), having 3BS, 5AS, 3BS + 5AS, and control plants without these QTLs [8,102,103,104]. The genotypes were inoculated by one *F. sporotrichioides*, one *Microdochium nivale*, two *F. graminearum*, two *F. culmorum* and two *F. avenaceum* isolates. Figure 2 shows the mean performance of the 24 genotypes for FDK as a percentage for the different *Fusarium* species and isolates, including *M. nivale*. The no-QTL group has the highest bars for all eight isolates. The 3BS and 5AS QTL mean values are very similar for all isolates [104], proving that the two QTLs have very similar strength in controlling DI, FDK, and DON. Further, the two QTLs together are positioned at the lowest bars, showing the transgressive segregation mentioned by Liu and Wang [105] for Sumai 3 (CM82036 comes from the Sumai-3/Thunderbird cross). As *M. nivale* responded similarly to how it did in the van Eeuwijk et al. [91] paper, it seems that the resistance to *M. nivale* is common with the resistance to other *Fusarium* spp.

In conclusion, the data show that a race non-specific resistance determines FHB resistance, and it seems that it is also species non-specific. This means that a higher resistance not only protects against *F. graminearum* but can also be effective against the other *Fusarium* species and *M. nivale*. This is important because high susceptibility to *F. graminearum* would also mean a higher susceptibility to the other *Fusarium* spp. with lower aggressiveness. Therefore, a breeding program is enough against *F. graminearum*, and this would also mean automatically higher resistance to the other *Fusarium* spp. In relation to food safety, the very susceptible genotypes can be similarly susceptible to other members of the *Fusarium* genus. Therefore, they are strongly exposed to their mycotoxin contamination. This supports the idea that multitoxin contamination is dangerous and possible in the susceptible variety group, which might suffer heavy infection from the less aggressive FHB pathogens that could infect highly susceptible genotypes without any setbacks. Ten years ago, such an idea would not have been feasible as the analytical side could not provide the background. Moreover, breeders were not aware of the problem because *F. graminearum* is also an unsolved problem. But this is also true now. The increasing number of multitoxin analyses in wheat could, therefore, be a good platform to support or disprove the idea. I think the research should test the basic problems with the best possible methodical background, and the use of the data and knowledge may evaluate simplified, less costly applications that support the breeding work. For this, in the last decade, high-throughput multitoxin methods have been developed to enable the necessary research (MycoFoss, RidaQuick, etc.). The many occurring toxins have another food and feed safety aspect. The interactions between toxins are less studied, so the real risk of the given grain based on DON contamination can be problematic and misleading.

The knowledge of these important fields is narrow. The many *Fusarium* spp. whose larger part has low pathogenicity are not considered as seriously as they should be. We should have further support for the idea that the resistance is not species-specific. When the QTLs affect a species-non-specific one, this applies only to the two QTLs tested (*fhb1*, *fhb5*). None of the other QTLs were investigated until now in this respect. We do not have so many data that could verify the general validity of the statement on a much wider basis than we have now. What is the situation for the others? As food safety is a general problem in wheat, multitoxin technology can provide a vast amount of data to reveal the scale of the problem and aid basic research in clarifying it. We would need more research and QTL analysis of a population exposed to three to five different Fusarium species to reveal the possible identity of QTLs for them. We would need research from different continents and regions to examine how the system is working to gain a higher understanding of it and provide better solutions. And we should be able to identify the genes behind the QTLs, which is currently a research gap. Of course, the pyramiding of the most useful QTLs is required to ensure the best possible resistance to disease and toxin accumulation [106], but it would be better to work with cloned resistance genes.

## 7. What Is FHB Resistance?

Arthur [107] first reported resistance differences, meaning stable and different infection severities between genotypes. There is also the problem of emerging mycotoxins and the food safety complex of the many *Fusarium* spp. present a considerable challenge. From the literature, it seems that the resistance types would be independent. The type could be understood as self-supporting variables without a closer connection between them that should be combined to have resistance. By way of definition, classic Type 2 resistance is clearly resistance to spreading, and it can be tested by following a point inoculation in the middle of the head and the spreading, e.g., the number of visually infected spikelets are counted several times or on a given day (21) after inoculation [108]. Type 1 is not well defined; Schroeder and Christensen [9] characterize it as resistance to infection (invasion) but consider it a result of the spraying inoculation and measure it by the number of infected spikelets. In practice, it appears that two traits were examined [109,110]. The incidence means the number of heads having one or more infected spikelets. The severity means the proportion of infected spikelets on a single ear or in a smaller or larger head population, which is estimated (counted) as a percentage; this has become accepted as a way to characterize it, and it has been described as the reaction of the plant to spraying inoculation. Their multiplication produced the FHB disease index. 

It is certain that Schroeder and Christensen [9] did not follow this testing methodology. Type 1 can be understood as the way in which spores from natural inoculation land on the outer surface of the lemma and palea before reaching the ovarium. Kang and Buchenauer [111] reported on the details of infection over the first few days. However, evaluating Type 1 resistance by measuring the number of infected spikelets means that Type 2 resistance is also included in this because to have visual symptoms on the spikelets suggests that it starts theoretically from the inoculation of the ovarium. Therefore, in the disease development, the two types overlap, making their differentiation impossible. Schroeder and Christensen [9] noted that the FDK can be a resistance-differentiating agent, recognizing its importance. In Type 2-resistant genotypes (Frontana and Erythrospermum), no FDK was found; for the rest, variable results were recorded. More recent research [103] defined Frontana Type 1 resistance. I cannot decide from the present facts and experimental material which is true. Schroeder and Christensen [9] entitled a subchapter “*Scab resistance as measured by the number of infected seeds*”. This was forgotten, but this was maybe the first occasion when FDK was used for resistance evaluation against FHB. Of course, the FDK as a phenomenon has been known for about 100 years [112]. The rediscovery of the significance of FDK [7,12] and the description of three further resistance types (resistance to kernel infection, the resistance to DON accumulation, and tolerance, meaning better yield at the same infection severity) are connected to the detection of DON for wheat breeding and searching traits that correlate better with DON contamination [7,94]. We can also list anther extrusion; the case is well documented [13]. As this is considered to participate in resistance forming, and the resistance components are not independent variables, its classification for us as to which resistance component it belongs is a secondary problem. However, during the selection process we can easily recognize and apply it.

The infection starts with the landing of spores (ascospores, conidia, and mycelium pieces) on the palea and lemma of the wheat head [113]. In humid weather, the spores start to germinate, and the germ tube at the edge of the lemma turns to the area between the lemma and palea and starts to grow, further attacking the inner surface of the palea. In humid weather, the orange-colored sporodochia develop on this edge with the mass of macroconidia of *F. graminearum* or other *Fusarium* spp. In this phase, the anthers that remain full or partly in the spike pose less risk because the pollen contains choline and betaine [114,115,116,117], and in wet conditions, the water extracts of these water-soluble compounds can indicate a much stronger growth of the *Fusarium graminearum* fungus (this is perhaps true for the other *Fusarium* spp. as well). The effect is not absolute: on more highly resistant genotypes, the spread of the infection and DON accumulation is much more restricted. It seems that the strong anther extrusion is riskier for susceptible genotypes. The growing mycelium has to grow into the embryo. From germination to the embryo is a long way, which is summarized in the Type 1 category and may consist of a number of physiological processes. It is very complex. Following an infection, many hundreds or thousands of genes become up- or down-regulated. When we test Type 2, we spare the long way to the embryo; Type 2 resistance starts in this phase. As the way is spared from spore landing to reach the embryo, the symptom development is faster in Type 2. Nonetheless, we can only count the diseased spikelets, as did Schroeder and Christensen [9].

What is known about the relationship between Type 1 and Type 2 resistance? Actually, not much is known. As they are tested with different methodologies, the results of the two tests cannot be compared, and in the background, the same QTLs can be expressed differently. Therefore, a test was made in the framework of the FP5 project (Fucomyr 2000–2003). Buerstmayr and Ruckenbauer developed the spring wheat CM80036/Remus population in Tulln and analyzed both Qfhs.ndsu-3BS (*fhb1*) (3BS Type 2) and *Qfhs.ifa- 5A* (5AS Type 1). QTLs were mapped by Buerstmayr et al. [102,103] in separate tests. From the mapping population, 24–24 DH genotypes were selected for the presence of 3BS, 5AS and 3BS + 5AS and the absence of QTL [8,104]. The test was performed by spraying inoculation [92,94]. The test results showed a similar influence of the QTLs for the three traits’ disease index, FDK and DON and this was validated in many independent tests [104,109,110]. The difference is that the 3BS QTL was less effective in controlling disease index, but in FDK, the responses were much closer, where all 3BS genotypes were slightly more susceptible than 5A plants. The resistance to DON accumulation was better for 3BS QTL than that of the 5A plants. This can be explained by transforming DON into non-toxic DON-3-glucoside that reduces DON content to some extent and improves resistance responses as DON is a virulence factor for the fungus, e.g., less DON production due to conversion to atoxic masked glycoside form reduces disease severity [101]. In this way, the more susceptible 3BS plants could be balanced by more efficient DON transformation. Other minor genetic effects can also be considered. Of these, only the DON data are presented in Figure 3.

The resistance identified following spraying inoculation was fully expressed for both resistance types. This means that all Type 1 resistance test results by spraying inoculation contain a Type 2 influence. They have equal significance; they share about 50–50% of the CM82036 resistance. From other papers, other QTLs are known from Sumai 3 (as they were not tested, more detail is not provided), but the highly different genotype differences within the individual group support this conclusion. This is the most important lesson. With this testing methodology, the effect of the QTLs could be correctly compared. This means that all data to characterize the two QTLs came from the same experiment. The results agree fully with the finding of Liu and Wang [105] that *fhb1* and *fhb5* (3BS and 5A) alone have a moderately susceptible resistance type, and on their list, Sumai 3 is among the best varieties with an extremely high transgressive segregation, e.g., the two QTLs can secure a much higher level of resistance than the superior QTL alone. Their moderate susceptibility was also proved in this test. The differences between resistance of the lines within the group are highly significant, e.g., further QTLs might be present in the population. What can be the real genetic value of resistance to DON accumulation? When we make a mean from the five most susceptible lines in each group, the data show medium susceptibility for both QTLs. This explains why the strong selection for *fhb1* resistance did not make the progress expected. In fact, it was expected that all the resistance of Sumai 3 was due to the 3BS QTL, when in reality, many of the 3BS plants were moderately susceptible according to their genetic background, but it was not considered that 5A was also responsible for the good performance. On the other side, *fhb1* QTL is characterized as the most important and most stable for FHB resistance [118]. I agree with its stability, but its significance seems to me to be far overestimated. I think that this is a very important source of the slow progress of the FHB breeding programs built on the QTL.

Spraying inoculation ostensibly allows for analysis of all resistance types described to date. Therefore, this can be called total or overall resistance [8,119]. This is what breeders require. With Type 2 resistance alone, we might overestimate the resistance in the varieties, by 50% in our case. Additionally, Type 1 resistance also has a spreading function at the population level and also within ear overlapping with Type 2. As an epidemic develops, the rate of infected heads rises also. The AUDPC analyses describe this for both resistance types [109,110].

We found this result for the two QTLs having high importance. But there is no information about the other QTLs. The mapping populations that were mapped with both spraying and point inoculation were considered, and we could select the necessary plants for the spraying inoculation test.

The QTLs may have different functions; for FDK and DI, this was confirmed also for the Frontana/Remus population with QTLs specifically for DI (3) or FDK (4), or both (3) [120]. In the Frontana/Mini Mano population, they influenced the three traits differently, as was found in the Mini Mano/Frontana DH mapping population [10], but here also DON responses were mapped. So, the differences between DI, FDK and DON responses presented all possible combinations in QTL functions, and from the 15 QTLs 4 were identified as having an influence on all three traits as it was proved for *fhb1* and *fhb5*. The DI, FDK and DON data must not agree, and the ecology and other influences can also cause further differences. Newer data fully support these findings [121], and genome wide studies also support the different regulation of the main FHB traits [122], in addition a part of QTLs seem to be common. QTLs with different regulation were also detected by Bonin and Kolb [123] for DON, but they were week explaining 4–6% of phenotypic variation. The three FDK specific QTLs varied between 7 and 12.3%, indicating the same conclusion as ours at much higher explained variance (20% or higher for the best QTLs [10]. In addition to the QTLs, there are unknown genetic paths resulting in DON overproduction or relative DON resistance detected, an important phenomenon that needs understanding and utilizing [89,95]. Problems with the DON gene cluster can also cause lower or no DON production for special isolates. However, in about 80% of the tested genotypes (the rate varies between tests), the resistance and DON contamination correlate well when testing one cultivar with many isolates. Vica versa, this is not true because of the different genetic habit of the varieties.

The synthesis of DON starts before the symptoms can be seen in a very early stage of the disease development [111,124], with DON identified 36 h after inoculation and its location established in the floret. Another source of the inconsistency between visual symptoms and the DON contamination level is that the last visual head blight rating finishes on the 21st day after inoculation at flowering [108] or in later genotypes, 2–4 days later. From this time to harvest, about three to four weeks remain. In dry weather, only a slight increase in disease development can be recorded; a longer rainy period, however, speeds up disease and toxin development. In such years, we found that 50% of the selected plants based on DI should have been discarded after FDK evaluation. Additionally, this also increases DON contamination. However, not only DON but also other toxins may accompany it and increase the health risks for humans and animals. A susceptible variety might cause a much wider toxin contamination than the more or highly resistant genotypes do, so DON alone will not be adequate for testing. We think that this will be a hot topic in future to observe and monitor the real risks that mycotoxins may pose. The role of plant breeding should be clarified, and practices should be established to further reduce multi-mycotoxin contamination.

The mycotoxins cause two additional problems. One is the presence of 13-acetyl-DON and 15-acetyl-DON, both of which are toxic, with 13-AcDON being more toxic. When only DON is measured, these two other variants are not considered, and, therefore, their toxicity might be significantly higher than that of DON alone. Their toxicity should be expressed in a DON-equivalent amount, as this would be more correct than the use of a single DON concentration. Another concern is raised by the masked mycotoxin problem, which refers to all mycotoxins, including DON [37]. The control of many toxins of the *Fusarium* spp. is a serious task and needs a nuanced approach.

In this paper, I do not intend to analyze the present position of QTL research. The main reason is that the basic system of the resistance components has a fundamental contradiction: the spraying inoculation method also shows Type 2 resistance well, so many of the QTLs identified by spraying inoculation can be Type 1 or Type 2. It was generally assumed that all QTLs identified by spraying inoculation are Type 1. It seems to me that this is not true. Therefore, all important QTLs should be retested to have a scientifically solid basis. At present, I believe that the spraying methodology will win for two reasons. It provides information on all resistance types that we need to combine. It is easy to handle and can provide 15–20 g grain following inoculation for many additional tests, which is problematic in the case of point inoculation. This inoculation technology is also used also for fungicide tests and in many other research problems where artificial inoculation is needed.

Plant height influences FHB response, taller plants normally have less severe symptoms. Draeger et al. [125] found, in Arina, that the Rht-D1 gene co-localized with a QTL on chromosome 4D also influenced FHB resistance expression when the plant height for the plants with and without a dwarfing gene did not differ significantly from each other. Rht-D1b (Rht-2) reduced plant height by 24 cm but doubled the severity of the FHB infection. However, *fhb1* (3BS QTL) and *fhb5* (5AS QTL) reduced FHB susceptibility by 6.5 and 11.5%, but *fhb5* alone reduced susceptibility by the same amount as Rht-D1b (Rht-2) increased it [126]. It is no accident that it is hard to breed dwarf and FHB-resistant wheat plants, and this is also a challenge for breeders. The *fhb5* might make a significant contribution to this work.

In the Frontana/Remus population [120], a long flowering period of 17 days was found, which means four to five inoculation occasions in a season. We presented data from two years of this experiment [8], where in the first year, the beginning of the field inoculation was warm, and at the end, it was cool; in the second year, it was the opposite. When we have two years with similar weather patterns, we can say that earlier genotypes are more resistant than late ones, or vice versa, at least according to the statistics. However, it is nearly impossible to ensure stable weather conditions in the field for this time of year and an additional 4–5 days with the same ecology. Another observation was that the FDK was much more stable than visual symptoms. To the best of my knowledge, this problem has never been discussed in papers dealing with QTLs for FHB. When the inoculation is administered in a controlled environment, such a conclusion could be drawn for the role of the flowering time, otherwise, the case needs special care. Most of the disadvantages of the QTL analysis stem from this problem.

Dweba et al. [127] concluded that breeding for FHB resistance was less successful than anticipated. The reasons for this, they suggest, are that the complementary gene pool is poor for exploring and manipulating genetic resources, the polygenic nature of the resistance hampers progress, and the low amount of information on pathogenesis, which could help to detect paths to make progress, is currently unknown now, or there is only limited information available at present. They mention, among other genomic technologies, more extended international collaboration, and the deployment of FHB-resistant genotypes for the grower. Here, I mention that there is a serious bottleneck in exploiting present knowledge. Most breeding sites have only one to two breeders for everything, and they do not have extra capacity to breed against FHB resistance, other than the many other traits needed. Plant breeding is, even today a mostly practical experimental science that lacks a strong theoretical background. Breeders require simple methods, by which a relatively large amount of breeding material could be screened, with a solid scientific background in state or foundational form to help them to utilize the best technologies at a reasonably low price. This is a task to solve.

The problem of resistance-related metabolites is also a prospective research target, but readily applicable results for the breeders are not yet in sight [127,128].

The other line with a much longer tradition is QTL research. Only one QTL is known (*fhb1*) with a postulated biological function [129]. In 2008, it was supposed that the required gene would be identified within months [130]. Su et al. [131] identified a hypothetical protein, TaHRC, that is probably responsible for resistance in *fhb1* QTL, but a final and validated result is not yet available. For biological function, it seems that FHB QTL *Qfsh.ifa-5A* connects to genes involved in cell wall biogenesis and terpene metabolism and a stress-response NST/1-like protein was identified as a candidate gene. Its function is not yet known, but the authors believe that it plays a role in anther extrusion [110].

When we consider the disease development and the epidemic background, **the different resistance types are rather resistance components, not independent variables.** During the disease development, the different metabolic pathways build on each other, strongly depending on the genetic background and environment. The many metabolic pathways contribute to the result we call resistance. They interact with each other and there are additional genetic mechanisms, for example, in the specific resistance to DON accumulation. Anther extrusion also belongs to this group, but an analysis is lacking for how this and other traits are genetically regulated and related.

As the relations are often variety specific, a general matrix does not seem to be the case. This background significantly inhibits the progress of the practical breeding work. I think that Dweba et al. [127] are right when they suggest that they require a better understanding of the physiology, pathophysiology and (I add) epidemiology of the disease process. These also determine the methodology that is inadequate in many respects (see Section 7). The basic methodical problem is the poor and simplified methodology that leads to more problems in understanding the nature of resistance.

## 8. Phenotyping, How to Evaluate Resistance

This chapter contains information from the isolation and identification of the fungi, evaluation of artificial inoculation methods, and evaluating and identifying the traits that are important to characterize the resistance and influence resistance expression essential. As FHB is a polygenic trait, it needs many data to determine its stability and usefulness and it is more environment dependent than resistances with mono- or oligogenic backgrounds like rusts. In this respect the situation is similar to evaluating grain yielding ability, where more locations and years are needed. This basic feature will not change, as it is better when we adapt to this situation. We screened several hundred papers, and we cited about 100 papers [87,88] with the conclusion that, in all cases, one inoculum was used and in about 80% of the papers only visual symptoms were tested, and mostly Type 2 resistance was analyzed. FDK or toxin was analyzed seldom.

We concluded that resistance is a complex mix of traits differing in the genotypes tested. For this reason, it is not possible to select any of them as a generally valid one for all cultivars. As food safety risk evaluation must consider toxin reaction. The first question is as following, which field trait correlates best with toxin contamination? Across many tests, the answer is that the FDK contamination is significantly closer to that of DON than the disease index is [7,8,87,89,90,92,94,95,132,133]. The disease index correlates with DON content at about an r = 0.2 difference lower than FDK does; even in several experiments, they behaved very similarly. As Schroeder and Christensen [9] found that FDK is a useful additional trait to evaluate FHB resistance, our data supported its significance: FDK is important for all inoculation methods. The FDK correlates well for a significant part of the tested genotypes with the visual data (less infection, less FDK), but this is not valid for the smaller, but significant, part of the genotypes. As toxin accumulation does not depend alone on resistance level (DI and FDK), without toxin contamination data, nothing can be said about the food safety risk of the genotype tested. When we screened the literature, in about 5–10% of the papers published FDK and DON data. Their level was often very low being only moderately suitable for the differentiation of the genotypes [8,89,134]. Despite many excellent papers like Zhang et al. [135] that analyze only visual symptoms, presenting important interactions of pyramided QTLs, toxin data do not support the conclusion. A recent paper found the FDK has the closest correlation trait to DON [136] and found with a sophisticated neural network technology correlation among FDK and DON between 0.41 and 0.58. Our simple visual data showed much closer relations [87,89,90,94,95,133]. It is considerable that recent studies are starting to agree with the higher significance of the FDK and DON in breeding programs instead of using visual symptoms [137,138].

This means that most of the papers deal with visual head blight symptoms. The assumption that the visual symptoms are suitable for forecasting FDK and DON could not be supported. Although, in a significant number of genotypes, the correlations between visual symptoms and FDK and DON are close, it is not a natural low. For 1% DI, where 96 data were behind each genotype mean, 1% DI DON varied between 0.40 and 1.56 mg/kg (minimum and maximum), for FDK 0.37 and 1.56 mg/kg DON (minimum and maximum), respectively. This means that about four-times as many differences were found [95]. In other tests, ten-times as many differences were recorded (year 2011) [133]. Finally, when a farmer sells grain, nobody asks about the field disease index or FDK, but the DON (and other toxins) value is decisive. I think, in addition to many scientific arguments, that this practical argument has weight. The disease index is often used for head symptoms. It can be evaluated directly as a percentage of the diseased spikelets. For Type 2 resistance, this is common. For Type 1, incidence and severity are normally used separately, their multiplication is identical to the disease index [19]. As in most cases, the incidence and severity are not connected to toxin evaluation, and their ability to decrease toxin contamination is less understood. As the disease index is generally used without any problem, as in the case of Schroeder and Christensen [9], I think that its use will become normal in future. There is, however, one problem that should be clarified: the evaluation of spikelet infection. There are spikelets showing initial infection when water-soaked lesions appear, which should be evaluated as infected. On the other hand, the whole spikelet can be destroyed and killed, and this is also one infected unit. Without FDK, the problem is hard to solve, but the evaluation of FDK can balance these differences. In some genotypes, we found perfect-looking grains with a powdery discoloration on the grain surface. In addition, the classic scabby grains, we also rated these ones, the DON analysis was made, and the correlation coefficient became 0.1 closer with powdery-looking grains than without them. They were probably subject to late infection. As these participate in DON contamination, their consideration is a reasonable decision.

Which inoculation method is better? It seems that the spraying inoculation covers all resistance types, so the total amount of resistance required can be achieved by following this methodology. Type 2 covers only a part; therefore, the amount of resistance can be overestimated. As most of the European good resistant materials were not mapped, for QTLs, the mostly spring wheat-originated QTLs detected do not provide any information about the winter wheat resistance level, even common QTLs could be present.

We used the spraying of a group of heads by Fusarium suspensions at about 15 heads from all sides to secure inoculation for each spikelet. Then, they were covered by plastic bags for 24 h. In 1985, a test with 21 genotypes with 24 and 48 hours’ incubation was employed [139] the mean disease index increased by one-third, and the yield compared to the 24 h’s incubation was also reduced by one-third (ecological conditions favored FHB). Then, genotype reactions differed: some kept their 24 h incubation. response at 48 h; for others, it was doubled. As the severity was high enough to demonstrate, we did not modify the 24-h incubation. The authors of [124] made a similar test with eight genotypes and did not find any difference between responses of the genotypes between 24 and 48 h incubation times. This result did not prompt us to change our approach. Later, in 1996–2000, following drier and warmer flowering periods, the infection severity decreased, and the 24 hours’ incubation was not enough; therefore, the incubation time was increased to 48 h, which served well for decades. The dependency of the variety resistance differentiation on aggressiveness is clear [8,21,61,94,95]. Therefore, for ensuring aggressiveness, the direct test is much better than any artificially set conidium concentration [140,141]. We should add that the mycelium is as good an infection agent as the mycelium [142]. This means that a conidium and mycelium mixture could equally produce a good performance.

Three inoculation methods were compared between 2009 and 2012 [95]; a summary is presented in Table 4. The year 2009 was extremely dry and hot, which strongly inhibited the disease development, in spite of the bag cover. We left these data in the paper, because they show that even spraying + PE (polyethylene bag cover) is the most effective approach; in some years, the results were unsatisfactory, even when we used four isolates. The DI had the most variable performance between years, but the four years for the spraying + mist irrigated version gave results 50% less than those for the polyethylene-covered bags. The maize debris showed the highest inconsistency and the highest mean DI; in two years, the variety differences were not significant. In FDK, the PE bag cover resulted in 2.5-fold higher FDK than the mist irrigation, and the corn debris was the least effective. Except for one year, both spraying inoculations allowed for, excellent, good or acceptable differentiation, whereas the plant debris inoculation was only twice acceptable. The largest difference was found in DON contamination, where the spraying + PE caused a more, than three-fold higher DON than the mist-irrigated version. The corn debris reached only 50% of the mist-irrigated version. This experiment showed that under dry and warm conditions, spraying inoculation with PE cover was the best of the methodical variants tested.

For research purposes, the existing literature applies highly specialized methodologies with enzymes causing resistance in different genotypes. The stability of aggressiveness lasts for a shorter time, being mostly stable for several years [143] but potentially changing after a longer period. The *F. culmorum* isolate No. 12551 and *F. graminearum* isolate No. 12377 were used in our tests for more than 20 years, and later, they should be changed, despite the suggested passage through artificial inoculation [45]. It is also the case that parallel inocula from the same test tube of an isolate could provide highly significant differences in aggressiveness [21,144]. Therefore, the aggressiveness of the inoculum is a necessity. In a seedling test, we found significant differences between different dilution rates. In some cases, the dilution ceased to be aggressive at a 10-fold rate; in another isolate, a 20-fold dilution caused no change in aggressiveness. The same happened with the mixing of the isolates; the result could not be predicted [145]. Both were retested in the field [87,88,140,145]. In addition to the visual head symptoms FDK and DON were also tested, and all three traits showed the same low rate forecast we found in the seedling tests.

For dilution [88], a ratio of 1:1 decreased DI by only 5–10% compared to the undiluted isolate. Larger dilutions caused stronger reductions in aggressiveness, but the variation between isolates was considerable. We found a similar situation for DON contamination. The reduction was largest for the most resistant genotype F569/Kö/F 569//Ttj/RC103/3/-Várkony/4/Ttj/RC103/3/81.60/NB//Kö, with Nobeoka Bozu (NB) at the same level with a reduction of 65–70% DON. The mean values across genotypes correlated excellently with DI, FDK and DON, showing a general tendency that is not necessarily true for the individual genotypes. This means that when we take any dilution ratio, it will influence the variety ranking. In conclusion, the smallest problem was caused when no dilution or a maximum 1:1 dilution was made. From this perspective, an absolute resistance ranking is hardly possible as genotypes react differently to dilution. Of course, this means that large differences remain stable, even their extent varies [88]. Since 1985, the aggressiveness of the inocula is always checked before use, especially as we found a close correlation between the seedling and FHB test results [140].

Many papers report the use of mixtures of between 2 and 20–30 isolates [87]. For mixtures, only 10 of the 55 papers contained DON data,11 contained FDK data, and 9 had all three. In our test, four isolates were used in all possible combinations, with 15 different inocula with and without mixing. Examining the DON data, the mean of the inocula was 50 mg/kg, and the arithmetical mean of the participating aggressiveness was 40 mg/kg. This means that the mixture overbalances the aggressiveness of the participating aggressiveness means, from a −17% decrease to a 208% increase. However, in every case, these mixtures were better where isolate 3 was included. The essential point is that any number of inocula can be mixed to achieve greater aggressiveness; there will only be one resultant aggressiveness that is not better or worse than an inoculum from a single isolate. Therefore, we do not think that a mixture would be superior to single isolates. So, we cannot forecast exactly what will happen; but, to some degree, an aggressive inoculum can balance a poorer aggressiveness. When no aggressiveness test is made, this balancing effect cannot be considered, and an unsuccessful experiment may be the case. It is more reasonable that low-aggressive inocula should be excluded. The conidium concentration alone is unable to regulate aggressiveness. Since no paper mentions this, authors simply use a conidium concentration they have read in another paper. To use a given conidium concentration means that dilution or concentration of the inocula is necessary, but its impact is unknown. This will decrease the reliability of the work.

For us, the use of more isolates (inocula) without mixing was an important consideration. This is closer to the natural condition, where every infection site is the result of the infection of a single conidium of millions in a field. When the specialized races are absent, the aggressiveness differences between isolates influence resistance differentiation; at higher aggressiveness, the differentiation is much better [12,95,143]. For this reason, with four isolates or inocula with differing levels of aggressiveness from medium to high, there is a higher probability of gaining a more reliable result. As the different isolates act under the same ecological niche, within the test, no environmental interaction is supposed from the ecology, and even the genotypes may react differently to the different isolates or aggressiveness levels. When a test takes two years in two locations with four isolates, this means 4 × 2 × 2 = 16 mean values for the four inocula in two years and two locations, when the test lasts three years (24 means of 72 data), this data set is extensive enough to count stability for the different genotypes and forecast a feasible risk for the variety registration or research. This is valid for DI, FDK, and DON. Of course, it is necessary to screen a natural infection and multitoxin stage, which provides feedback for the breeder and grower on the value of the cultivar grown.

When should we inoculate? The traditional answer is during flowering, also called the susceptibility window. For this question, no clear answer was found [86], but an experiment was performed with seven genotypes with differing levels of resistance and inoculated at full flowering and at 4, 8, 11, 13 and 16 days later. The visual symptoms did not show a significant decrease during this period, e.g., the infection of the outer surface of the glumes and paleas was as successful on the 16th day as at the first inoculation. However, in FDK, after 8 days, a strong reduction of about 60% was recorded and later an additional 33%. DON was the most sensitive: the first three inoculations gave a mean between 9.23 and 9.81 mg/kg; on the 11th day, 3.45 mg/kg was measured; the last two days showed only 1.52 and 1.11 mg/kg, respectively. The process was similar for all resistance classes tested. The susceptibility period lasts for about 8 days, with smaller changes in the actual data of the three traits. Therefore, in a mapping population or variety screening, 4–5 days flowering distance can be treated at the same time, thereby reducing the number of inoculations. The data from the different inoculation dates of the population tested can be pooled when the maximum and minimum values and their means are close (not significantly differing). When this is not the case, the data should be analyzed separately; otherwise, false positive or negative qualification can occur as a result. I have not seen any paper where this problem could have been discussed. In variety testing, the problem is reduced, because to each ripening group, control cultivars with differing resistance levels are added. In a mapping population the largest problem is the heterogeneity of the population for flowering time and plant height. In developing the Frontana/Mino Mano population, the extra early and late lines were omitted and the extra tall and dwarf plants also, so two inoculations with six independent isolates were sufficient, and in the two years of testing, the means for the two inoculations were similar; therefore, the data could be pooled without any correction risk [10]. As a result, the LOD values for the QTLs increased two- to three-fold and the best QTLs explained more than 20% of the variation compared to the best QTLs made according to the original methodology (8%) [120].

Summarizing the methodological lessons, the increase in incubation time from 24 to 48 h significantly increases resistance expressions for individual genotypes, but varieties may react differently. The different aggressiveness level of the diluted inocula may also change variety ranks, mostly in the neighboring genotypes on the line, but larger deviations also occur. The dilution or concentration has a similar effect, and the influence of the mixing cannot be precisely forecast. We know that FHB resistance is a polygenic trait, with several important symptoms having earlier or later gained importance. Therefore, to identify only one trait surely does not serve long-term breeding goals. This is possibly one cause of the slow progress in FHB breeding. A correct evaluation of the risk is possible when data for DI, FDK and DON response are fairly evaluated, and not only DON but also the other traits are helpful in understanding what happens; should we consider whether there is a DON contamination that is proportional to the symptoms, or whether we should discuss DON overproduction or relative resistance? The other aspect is that a simple test will not inform us about the resistance capacity (amount of resistance) of the given cultivar. For this, we need a considerable amount of data. The standard methodology with one inoculum across two years is not enough to provide reliable data on the stability and amount of resistance. This is similar to the yielding ability tests, where, normally, 10 or more experiments are made annually in different locations to identify genotypes that adapt best to different conditions. Therefore, the FHB tests must follow this way of thinking, and through this, we can identify both the superior and the good adaptable genotypes suitable for successful growth under varying environmental conditions, for breeding, variety registration and genetic research.

## 9. Breeding Aspects

The breeding for FHB resistance is possible. Figure 4 shows the differences from the Szeged FHB breeding program. In the SS nearly no healthy grain is seen, in the R 1–2 infected grains can be identified. For us, therefore, not the FHB resistance is not a problem by itself, but its combination with other resistances and quality traits is. This work is in progress with a good outlook.

We have to differentiate between the special methodologies applied in genetic research, identifying QTLs, resistance genes, their functions that differ from QTL to QTL and from gene to gene, their interactions, etc. This is important to develop more resistant germplasms and varieties. It is known that the FHB resistance itself does not cause a reduction in yielding ability, e.g., in all resistance classes we can identify high yielding genotypes [16]. We have similar unpublished results fully supporting this view. In the whole cereal literature only one paper was found that could prove yield penalty caused by a resistance gene in barley [146]. For other cases, the poor yield of resistant material comes from the poor agronomy performance of the resistance donor landraces, related species or wild relatives.

In the serial breeding work, the first important task is the large-throughput screening work that can identify superior genotypes present in every nursery. As much less work has been conducted on winter wheat, it is very important to detect the medium-resistant materials and utilize them [89]. The crossing partners should have at least medium resistance to stand a good chance of achieving similar or higher resistance. The F_1_ generation should be checked for FHB (DI and FDK), and the susceptible combinations should be discarded. In this case, the F_2_ and F_3_ generation should not be inoculated, but otherwise, it is necessary. The first toxin test is due when the first 5 m^2^ yield trials are made but only for the genotypes that yield at control level or higher and the DI and FDK are lower than average. Lower yielding lines with otherwise good or excellent resistance should be retained for crossing and scientific purposes as germplasms.

However, we require a methodology that also allows for the possible exact determination of the food and feed safety risk of a variety even in cases when no genetic information exists. In winter wheat breeding, the overwhelming majority of the breeding material is unknown for FHB resistance. Therefore, resistance screening is the first task. In this work, the determination of the resistance level to the disease and resistance to toxin accumulation is decisive. As a rather large variation exists, the detection of higher resistant genotypes is possible [89]. It has been the general experience for decades that only a careful screening of the breeding material can produce results.

What traits should be checked?

Based on the visual symptoms, the FHB resistance (disease index DI) should be evaluated.As DI and FDK are often differentially regulated, and DON contamination correlates significantly more closely with FDK, its evaluation is important. The results were confirmed by Wu et al. [136]. They evaluated a highly sophisticated neural network methodology to evaluate FDK to improve the accuracy of genomic selection. As phenotypic and genetic correlations were lower, we achieved up to r = 0.80 or better in some tests [89], though we do not feel that this method should be changed.The genetic regulation of DON is often different to that of the DI and FDK, and the official limits apply to the toxins (DON, etc.), while the food safety risk cannot be evaluated without toxin measurements. There are qPCR markers to measure fungal mass in the wheat tissue for different *Fusarium* spp.; therefore, a multispecies approach is possible. As DON contamination for one percent FDK showed ten-fold or higher differences, a forecast of the toxin contamination on genotype level is problematic; even closeness of the correlation is medium and significant. Further research is necessary to see clearly see the situation we face.

Based on these, the evaluation of toxin overproduction, relative toxin resistance, toxin production for 1% DI and FDK can be counted [8].

As genotype responses to the different isolates of *F. graminearum* may respond differently, four independent isolates are needed to provide a good overview. As the literature stresses the ecological dependence of FHB and its traits in breeding practice, two years seem to be acceptable. For variety registration two locations and two years are suggested; and when the regular tests are planned for three years, the FHB tests also last three years. For three years and two locations, 24 mean data (of three replicates) serve for ANOVA to measure stability and other indices that characterize the FHB risk of the genotypes.

The FHB resistance is race non-specific and probably species non-specific [91,92,94,134,147]. This forecasts stable resistance for a long period. The only danger is the selection of more aggressive isolates in epidemic centers. In the Chinese Yangtze valley, an increase in the more pathogenic isolates from the *F. graminearum* clade was observed [44,148], which can jeopardize resistance and low toxin contamination. This needs confirmation in the pathogenic population.

The multitoxin problem will highlight the problem of resistance to different *Fusarium* spp. It would be fruitful to organize an international consortium, to test again the problem and not only DON, but the major toxins of the other selected *Fusarium* species could be analyzed, too. I think that the increasing number will consider starting the scientific basis more carefully and wider than thought until now.

## 10. Influence of Resistance to Agronomic Traits

Resistance determines fungicide efficacy against Fusarium head blight [148,149,150,151]. The more resistant plants could be successfully protected with good fungicides that could not reduce the DON contamination below the EU limit in susceptible varieties. In this respect, Kleber et al.’s [152] results are important, suggesting that a fungicide spraying could successfully control deoxynivalenol, deoxynivalenol-3-glucoside, nivalenol, and nivalenol-3-glucoside at the same time. It also decreased the contamination by culmorin and its hydroxy metabolites 5- and 15-hydroxyculmorin, as well as aurofusarin, but also increased the concentration of *Alternaria* mycotoxins. The other mycotoxins were not tested, although their toxins may play an important role like enniatin A and B. This might be important in decreasing Fusarium mycotoxins in grains that the plants were not selected for. For this reason, this research area will have a higher significance in ensuring higher food safety. The whole system is rather complex, and careful research is needed. A more or highly resistant variety provides significantly less risk after sown corn or other plants having common pathogens with wheat, and not only for Fusarium but also for other diseases [148,149,150,151]. Kleber et al. [153] concluded that the toxin reduction was not full and that a fungicide treatment should be developed, and we can add that the highly sensitive cultivars cannot be protected successfully. Resistant plants will better tolerate tillage without plugging, or, only rarely, the soil loosening better serves the utilization of rain, and the high microbial activity of the soil ensures a much better decomposition of infected debris than we previously thought. Nakayima et al. [152] proved the significant increase of FHB in lodged stands; therefore, standability is important to reduce toxin contamination. The key for better protection is the main argument to introduce the official variety registration for FHB resistance and resistance to toxin accumulation. This is now ready to use and can significantly contribute to reduced losses in epidemic years. Growers should apply the breeding recommendations when an official qualification shows a low risk of the varieties having higher added value for commercial production.

## 11. Conclusions

Most of the literature focuses on FHB concentrates on *F. graminearum* in wheat and deals mostly with the visual symptoms. The existence of common resistance to different *Fusarium* spp. could be proven. However, this is not yet supported by other sources. In terms of food safety, this means that genotypes susceptible *to F. graminearum* are susceptible to other less-pathogenic species that produce highly poisonous toxins. As multitoxin methods are available everywhere, the problem should be tested with a hypothesis about how far resistance to *F. graminearum* could control other *Fusarium* spp. and their toxins. For this latter concern we do not have any data as far I know. Higher resistance is important not only because of combatting the disease directly but also because mycotoxins should be controlled. The fungicide effect also depends on the resistance level [1,133,134,149,151,154,155,156]. The careful and scrupulous phenotypic selection is the key to a successful breeding program. Buerstmayr [157] and Dill Macky [158] have the same opinion. When fungicides cannot reduce DON contamination in susceptible genotypes below a certain limit, these genotypes should be withdrawn from commercial production and replaced with a moderately or highly resistant variety that can manage it. Higher resistance also decreases the predisposition of plants to risky previous crops like maize, and the consequences are less harmful. As there is no immunity to FHB, other than higher resistance, careful management is necessary to protect the plants [159]. The significance of FHB resistance will increase, as the EU prefers agriculture with minimal pesticide use, increasing organic production where the use of traditional pesticides is not favored. The sharp increase in input prices also favors higher resistance where appropriate, not only to FHB but also to other diseases. The multitoxin contamination is a fact, and both the regulated and non-regulated toxins should be carefully researched to prepare for the challenges of the future [160]. This is necessary to increase the competitiveness of wheat production at home and abroad.

## Figures and Tables

**Figure 1 toxins-16-00031-f001:**
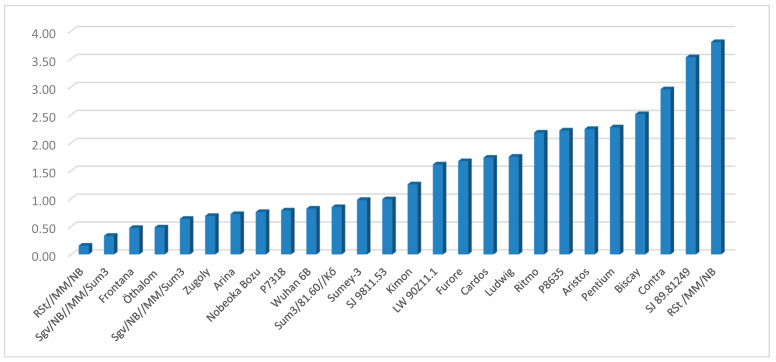
FHB resistance in wheat, DON production (mg/kg) for one percentage of FDK via infection severity, Szeged, 2001–2002 [94].

**Figure 2 toxins-16-00031-f002:**
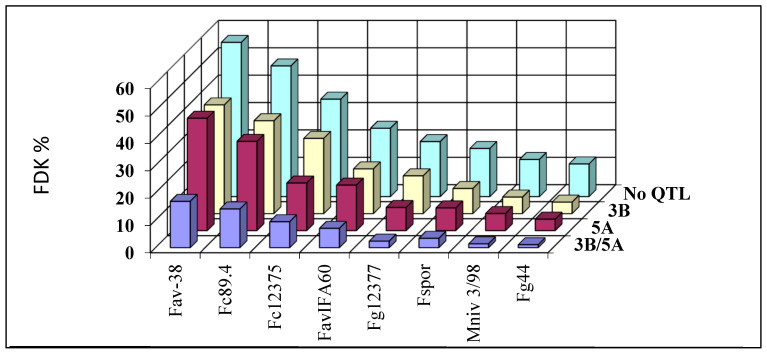
Resistance protection of QTLs 3BS and 5AS against eight isolates of Fusarium on the CM82036/Remus population in wheat, Szeged-Tulln, 2002–2003. Fav: *F. avenaceum*, Fc: *F. culmorum*, Fg: *F. graminearum*, Fspor: *F. sporotrichioides*, Mniv: *Microdochium nivale* [8]. The numbers after species designation are the isolate names.

**Figure 3 toxins-16-00031-f003:**
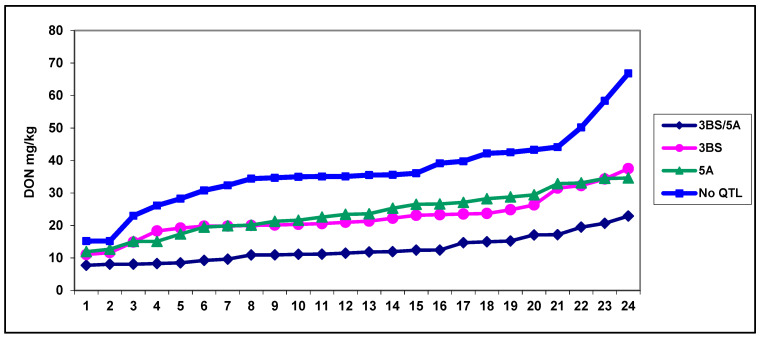
DON content (mg/kg) in the CM82036/Remus population across years, locations, and *F. graminearum* and *F. culmorum* isolates, 2002–2003. LSD 5% for genotypes 3.37, for QTL groups 2.16 [8,104].

**Figure 4 toxins-16-00031-f004:**
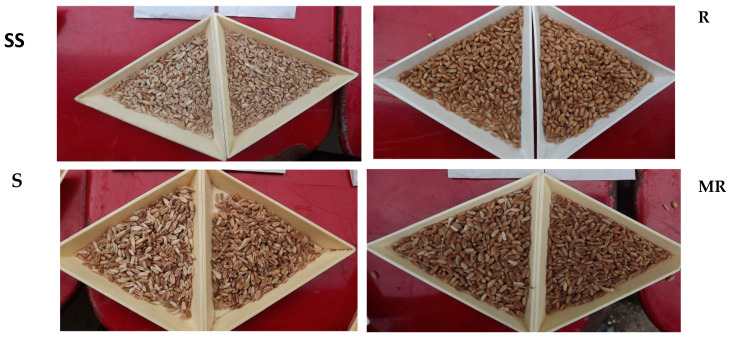
Resistance of wheat genotypes from the Szeged FHB breeding program, SS susceptible control, the 15 heads in bunches were inoculated by 2 different isolates. SS = super-susceptible, S = susceptible, MR = moderately resistant, R = highly resistant.

**Table 1 toxins-16-00031-t001:** Losses of wheat production chain, preharvest and postharvest combined, based on [30].

Item	MMT	% to Total Capacity	% to Harvested Yield
Total capacity	1161	100	
Total harvested	774	67	100
Preharvest Loss, Biot *, abiot., stresses, 33% including harvest loss 3%	387	33	-
Storage losses	154	13	20
Mycotoxin contamination	78	7	10
Consumer and other waste	101	9	13
Total waste	720	62	43
Total grain consumed	441	38	57

* Biotic and abiotic.

**Table 3 toxins-16-00031-t003:** Resistance of winter wheat genotypes to five *Fusarium* species, FDK, 2001–2002 [94].

Genotype			*Fusarium* species			Mean	Variance
	Fc	Fg	Fsamb	Fav	Fspor		
Sumai-3	8.58	0.75	2.17	0.08	0.75	2.50	12.3
Wuhan 6B	12.00	1.08	3.17	0.83	2.17	3.93	21.6
Sum3/81.60//Kő	8.92	1.38	12.92	2.46	1.00	4.49	28.3
Nobeoka Bozu	12.17	6.75	3.00	1.25	3.33	5.56	18.7
Sgv/NB//MM/Sum3	18.00	5.67	3.50	0.54	0.63	5.91	52.1
RSt//MM/NB	21.08	2.54	17.50	4.21	1.83	8.54	83.3
Frontana	31.25	6.38	15.83	0.17	0.46	10.26	170.8
Sgv/NB//MM/Sum3	32.08	24.08	30.83	13.83	2.08	19.44	159.3
RSt//MM/NB	55.83	37.50	29.17	4.50	2.96	25.64	506.4
Arina	64.17	33.33	24.17	4.42	2.08	25.80	637.8
Furore	70.67	49.17	45.83	14.71	1.79	35.39	773.6
P 7318	61.25	40.25	50.00	26.79	7.77	35.79	431.5
Ludwig	72.92	57.50	47.50	11.88	2.42	37.44	909.3
Zugoly	88.33	78.33	35.00	1.67	7.00	42.85	1591.9
Kimon	85.08	65.00	58.33	23.96	6.92	46.69	1009.2
Aristos	77.92	59.17	63.33	26.08	16.83	47.04	676.1
Öthalom	91.17	57.08	78.33	5.25	24.42	48.24	1299.2
P 8635	85.17	76.67	61.67	18.58	19.08	51.19	1000.5
Contra	92.50	76.67	65.00	17.38	11.58	51.25	1312.0
Ritmo	87.67	65.00	76.67	27.33	23.50	53.74	847.2
Carlos	85.75	72.08	64.17	36.25	18.79	54.44	745.8
LW 90.Z 11.1	88.42	68.75	73.33	52.17	5.75	55.94	1009.9
SJ981249	95.83	69.17	78.33	33.79	26.63	58.80	875.5
Biscay	96.08	87.08	77.50	40.21	14.25	61.42	1195.2
Pentium	92.92	72.92	81.67	45.67	43.00	65.63	488.2
SJ981153	98.17	82.50	90.00	26.75	47.54	66.66	929.9
Mean	62.84	46.03	45.73	16.95	11.33	35.56	645.60
***Fusarium* spp.**	**Fc**	**Fg**	**Fsamb**	**Fav**	**Fspor**		
Fg	0.975						
Ssamb	0.930	0.906					
Fav	0.701	0.717	0.811				
Fspor	0.660	0.651	0.788	0.585			
All significant at *p* = 0.001.							

Fg: F. graminearum, Fc: F. culmorum, Fsamb: F. sambucinum, Fav: F. avenaceum, Fspor: F. sporotrichioides.

**Table 4 toxins-16-00031-t004:** Comparison of two spraying and corn debris-mediated inoculation for DI. FDK and DON 2009–2012 [95].

Trait	Year	1. Spray + PE *	2. Spray + M **	3. Debris + M ***	Mean	LSD 5%
DI %	2009	2.72	5.17	0.09	2.66	
	2010	**13.61**	**12.81**	**26.91**	8.97	
	2011	**15.41**	**8.61**	**6.52**	10.18	
	2012	**14.18**	1.37	2.26	5.94	
	Mean	8.08	3.79	8.95	6.94	0.31
FDK%	2009	2.66	0.81	0.20	1.22	
	2010	**16.24**	**8.91**	**7.60**	10.92	
	2011	**27.1**	**13.95**	**7.19**	16.08	
	2012	**30.22**	**4.28**	3.15	12.55	
	Mean	19.06	6.99	4.54	10.20	1.66
DON ppm	2009	2.12	0.72	0.40	1.08	
	2010	**12.82**	**5.43**	1.56	6.60	
	2011	**33.66**	**9.55**	**4.79**	16.00	
	2012	**9.27**	1.62	1.82	4.24	
	Mean	14.47	4.33	2.14	6.98	0.67

Humidity: * Polyethylene bag, ** Mist irrigation, *** Maize debris and mist irrigation. Bold: Suitable for differentiation.

## Data Availability

As this is a review article, data are available in the cited papers.

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
