# Peer review of "What Is Fusarium Head Blight (FHB) Resistance and What Are Its Food Safety Risks in Wheat? Problems and Solutions—A Review"

_toxins, 2024, doi:10.3390/toxins16010031_

Round 1

Reviewer 1 Report

Comments and Suggestions for Authors

This is a comprehensive and lengthy review on wheat FHB resistance and related food safety issue. I could imagine that the author(s) must have spent a lot of time to work on it. Generally speaking, it is a nice review paper and I like it, and part of content (not all) are useful for wheat breeding. I would recommend publication after my following concerns are appropriately addressed.

 Major concerns:

1.       Please shorten the paper and focus the most important issues. Too long paper may keep the reader off;

2.       I may not agree with the authors on too much elaboration on type 1 resistance. From my opinion and based on my undertanding and experience (I am not pretty sure I am right), the author may confuse incidence with type 1. If type 1 resistance really exist, we may see immune phenotype to FHB, unfortunately, no evidence supports existence of type 1, So far.

3.       Anther extrusion does influence the incidence, but may not change the resistance/susceptibility nature of wheat, therefore the attempt of defining anther extrusion as a new type of resistance may be misleading.

4.       The author(s) also suggest FDK could be a signal trait for measuring DON accumulation, which is also suspicious. FDK should be a consequence of type 2 resistance, and may have a weak, at least not strong, connection with DON content.

5.       I do agree with the authors’opinion that precise phenotyping is crucial for both theoretical and breeding research on FHB resistance. However, the author strongly recommend usage of disease index and FDK for phenotyping. I do not think disease index and FDK are good traits for phenotyping FHB resistance.

6.       The authors should include more recently published papers on phenotyping, and too many repeated citations of one or two original papers may not be good enough for a review paper.

Minor points:

There are several typing errors or grammatical problems throughout the paper, for example:

1)      several “)” in the paper;

2)      P2: I do not understand“As research has established that the three variables can be different, and the QTLs might regulate them differently [10] higher resistance ˙expressed does not automatically guarantee low toxin contamination”. Please reformulate the sentence.

3)      P4: Please change “Jiangshu” to “Jiangsu”.

4)      P6: Suddenly introduce new term “aw”and “aw”, and also should keep consistency in spelling.

5)      P8:”closely; indeed, their resistance is ostensibly the [16,21,92,93]”, please reorganize the sentence;

6)      P10: “…causes only sporadic infection if at all”, I do not understand.

7)      P11:” In their review, [101] found that DON contamination of the grains in the resistant plants”, please correct the sentence.

8)      P20: trivial descriptions on experimental details are not recommended in a review paper. For example,“For dilution [146], a ratio of 1:1 decreased DI by only 5-10% compared to the undiluted isolate. The largest difference was found in a 1:4 dilution ratio with a 10-40 decrease in DI. The variety responses were rather different. At a 1:1 dilution ratio, the DI increased by 5-15%, with five genotypes showing different decreases up to 20%. At 1:16, the decrease in aggressiveness was between 28 and 44%. The FDK behaved differently: a 1:1 ratio caused a decrease across varieties of between 5 and 20%, and the largest difference between isolates was found at a 1:4 dilution ratio. At a 1:16 ratio, the decrease was between 30 and 45% compared to the rate without dilution. For the cultivar reaction, ranks of the cultivars differed at different dilution ratios. We found a similar situation for DON contamination”.

9)      P21: “…of inoculations. The data can be pooled form the different…”, in which “form” should be “from”?

10)   P22:”… more effective resistance against toxigenic fungi also during the breeding process. It an important aspect of the FHB resistance problem that the FHB resistance does not… “, what does that mean?

1)       P24: “The request for a careful and scrupulous phenotypic selection careful is the key…”. Please reorganize the sentence.

Comments on the Quality of English Language

the quality of English language is good

Reviewer 2 Report

Comments and Suggestions for Authors

The authors did a good job in reviewing losses due to FHB in wheat production, casing agents, mycotoxin contamination in stored wheat, multitoxin presence, resistance to Fusarium spp., component of FHB resistance,  resistance evaluation, breeding, influence of resistance to agromic traits, etc., and I have not found many problems in the scientific aspect. Below are some points I’d like to discuss with the authors.

Type I resistance discribes the resistance to initial infection and should be identified before the sympotum  spread to  adjacent spilelt. Spraying inoculation can be used for evaluation Type I resistance, however, if the evaluation time is several weeks after innoculation, the resistance inclues type1 and type2. Therfefore, when we evalute the resistance to FHB, it's better to distinguish the resistance type and the inoculation methods.

In genetic analysis and QTL mapping, type 2 is used to identify the FHB resistance in different materials because of its stability and accuracy. Which is also of application value in breeding, because the FHB severity of wheat before harvest is greatly affected by the FHB spread. For example, Sumai 3 is a type 2 resistance variety and has stable resistance to FHB in field.

In the evaluation of wheat FHB in breeding, DI, FDK and DON often have a high consistency, so it seems insufficient evidence that DI, FDK and FDK have a large difference.

Comments on the Quality of English Language

The language needs to be redinded.

Reviewer 3 Report

Comments and Suggestions for Authors

-This manuscript is very thoroughly written dealing with Fusarium head bight resistance, food safety risks and possibilities for breeding resistant cultivars in wheat. I have a few comments on it.

-The authors cover mycotoxins, disease index and scabby kernels, but I think that they should also include qPCR, which is measuring the amount of different Fusarium species in grains. It has been found that the amount of F. graminearum DNA in grains correlates very well with DON levels and there is also a good correlation with e.g. F. langsethiae/F. sporotrichioides DNA and T-2/HT-2 toxins and F.avenaceum DNA and enniatins/moniliformin.  The changes of Fusaria DNA levels in grains precede other markers of resistance, such as DI, FDK and DON and it is easier to measure Fusarium DNA levels in grains than DI, FDK and DON levels. Fusarium DNA levels can also be measured before harvesting and they can be used to predict DI, FDK and DON levels during harvesting.

-Table 2: The authors should include newer information about Fusarium species and their toxins after the year 2002 in Table 2 (Bottalico and Perrone 2002). F. graminearum has been divided into several species (O'Donnell et al. 2004), of which F, graminearum sensu stricto is the most common in Europe. F. avenaceum is also a species complex according to Laraba et al. and Yli-Mattila et al. (2022). According to the them and Jestoi et al. (2004) F. avenaceum is not important as a BEA producer, while F. tricinctum is also producing enniatins. A new species, which is the main producer of T2/HT2 toxins (F. langsethiae) in cereal grains in many European countries, has been found (Torp and Nirenberg, 2004, Yli-Mattila et al. 2004), but the authors do not tell anything about it.

There are also a lot of small linguistic errors, some of which I have marked in the text together with  few comments (attached).

Comments on the Quality of English Language

Author Response

Dear Reviewer 3,

Thank you very much for your opinion and remarks for to paper. They were very useful in improving the paper. Several were cited in other part of the paper. Your questions directed my attention to problems I interested less, but they are important. and closely connected to the multitoxin problem we face also in the blood and urine samples of humans and animals. So scientifically peripheric species receive a significance and gave more research task what to do with them. Resistance? Other practices? We need basic research first what is going on.  All this, whet the main problems for the so called major species are mostly unsolved.

Thank you again,

Yours sincerely

Comments and Suggestions for Authors

-This manuscript is very thoroughly written dealing with Fusarium head bight resistance, food safety risks and possibilities for breeding resistant cultivars in wheat. I have a few comments on it.

-The authors cover mycotoxins, disease index and scabby kernels, but I think that they should also include qPCR, which is measuring the amount of different Fusarium species in grains. It has been found that the amount of F. graminearum DNA in grains correlates very well with DON levels and there is also a good correlation with e.g. F. langsethiae/F. sporotrichioides DNA and T-2/HT-2 toxins and F.avenaceum DNA and enniatins/moniliformin.  The changes of Fusaria DNA levels in grains precede other markers of resistance, such as DI, FDK and DON and it is easier to measure Fusarium DNA levels in grains than DI, FDK and DON levels. Fusarium DNA levels can also be measured before harvesting and they can be used to predict DI, FDK and DON levels during harvesting.

Thank you, when we want to evaluate on a broader way than DON, we should analyse the until now secondary considered Fusarium spp. at a higher rate. This remark is important as I represent also this opinion and your comments help to have a wider basis for the paper.

-Table 2: The authors should include newer information about Fusarium species and their toxins after the year 2002 in Table 2 (Bottalico and Perrone 2002). F. graminearum has been divided into several species (O'Donnell et al. 2004), of which F, graminearum sensu stricto is the most common in Europe. F. avenaceum is also a species complex according to Laraba et al. and Yli-Mattila et al. (2022). According to the them and Jestoi et al. (2004) F. avenaceum is not important as a BEA producer, while F. tricinctum is also producing enniatins. A new species, which is the main producer of T2/HT2 toxins (F. langsethiae) in cereal grains in many European countries, has been found (Torp and Nirenberg, 2004, Yli-Mattila et al. 2004), but the authors do not tell anything about it.

Thanks for the comment. For Yli-Mattila I found a later article from 2010. It has the number [38] now. This later publication also contains the F. langsethiae story shortly. For Jestoi et al 2004 another paper was found from 2007 Uhlig et al. [39], O’Donnell et al 2004 was changed for a newer paper, O’Donnell et al. 2018 under [40]. as the 2004 paper was cited later at No [100]. It was simpler to write here a later publication, as the 2004 paper was more important there.  For the F. avenaceum story, Yli-Mattila et al. 2022 was included [41]..

For F. langsethiae I intended to give the authors of the species designation and not a citation. Therefore, I omitted the  (  ) marks to make clear it similarly to F. graminearum Schwabe where normally no year is written.

Page 7. The two sentences are omitted,  and replaced with the statement that F. tricinctum is an enniation producer.

Page 8. Fugal was changed to fungal.

Page 11 glucosyde was changed to gluciside.

Page 13. What about F. graminearum level in plant measured by qPCR. It correlates well with DON levels. Also F. langsethiae DNA levels correlate well with T27HT2 levels and F. avenaceum levels correlate well with enniatin and moniliformin levels. So, why not to measure FHB resistance by using qPCR?

We made two tests  with qPCR, and we received a correlation between pPCR and DON at r= 0.55 for F. culmorum isolate 12375, but the visual FDK gave r=0.75 with DON.  Since that time, we did not made new tests, this seemed us better and cheaper. The more important argument is that symptom severity, FDK and DON can be differently regulated (see answers   to Rev. 1 and 2) it is true.  So,  a significant part of the tested varieties or genotypes will respond differently from the supposed value in DON, in a correlation the rate of these genotypes can reach 50 %.  As in advance we have no information about these regulation on variety level, we cannot avoid the careful investigation. It is also important, because we need to decrease the toxin contamination and give warranty that from food safety aspect what is the risk value of a commercial variety.

,

What about the amount of F. graminearum DNA (or the amount of DNA of several Fusarium species), which can be measured by qPCR?

As in Hungary we have red wheats, the recognition of the FDK is not complicated. This is more exact than the PCR data. However, in Australia where white grain color preferred, the visual scoring of FDK is not feasible, we had 1-2 such genotypes in test, I evaluated it for zero and 8 ppm DON was found.  We have a similar problem when the preharvest time is rainy and the grain color becomes pale, the  FDK estimation on the full size grains will be problematic. In this case, a qPCR test may help to receive data on the infection.  According to the recent data the multitoxin presence seems to be a generally valid feature. Therefore, their identification can be an important task. As we have seen, the DON production for a percent of FDK can show 10fold or higher differences. I do not know whether this relies on the other Fusarium spp and their toxins. l would not be surprised when this would be the result. There are now multitoxin rapid tests, the MycoFoss instrument that can measure six toxins at the same time in 7-8 minutes.  A kit for 100 measurements for  six toxins costs 900 Euro e.g., 1.5 Euro per toxin data. This is a cheap alternative for natural contamination up to 30 mg/kg DON. In artificial inoculation this is less useful as the contamination for the different toxins can be very high. By this was I have direct toxin data. for the qPCR I have only a fungal mass, without knowing the toxin limit.  The DNA extraction is not cheap, the running of the PCR machines also takes money, so we can think what is better. As I see the Senatori et al. 2023  article, it has a reason to consider this updated methodology.

It is important that you ask for the multi Fusarium infection. We have isolated 16 species of Fusarium in Hungary. From a grain sample it is normal that several fusarium species will be isolated, but they are possibly from inoculation of different conidia. In several cases we isolates two Fusarium species aafter surface sterilization form the same grain. Therefpre we started to think what is the genetic background of the resistance to these Fusarium spp. We published several papers between 2002 and 2008 about this problem, a summary in 2020. However, nobody else published such data. When our results will be supported by other research group, would be a significant development. I think, the multitoxin problem will make researchers more sensitive to this problematic. When this fits, and the resistance to F. graminearum fits to other Fusarium species it means that fhb1 and fhb5 QTLs are race-non-specific and species-non-specific, it would mean, that the most severe multitoxin problem can be forecast at the susceptible and very susceptible cultivars, e.g., they are mostly jeopardized.  On the other side we can hope at very high resistance like Sumai 3 good resistance and low toxin contamination against the different Fusarium spp.  will be the result.  I think, such a research program would be necessary and also qPCR could be involved into the tests. As this is not a small project, an international  project for four years could support the breeding and research finding better solutions than we have now. I will add a short comment to this in the paper.

There are also a lot of small linguistic errors, some of which I have marked in the text together with  few comments (attached).

I looked all, I answered the suggestions above.

I have corrected them, comments see above and also in the text of the paper.

Reviewer 4 Report

Comments and Suggestions for Authors

I think that the present review needs further improvement before being considered for publication.

Section 1

Sentence 1, the author should fully write FHB: Fusarium Head Blight.

I can’t understand why some sentences are written in italic throughout the text.

I think that the authors should better explain the active and passive resistance.

“As research has established that the three variables can be different, and the QTLs might regulate them differently [10] higher resistance ˙expressed does not automatically guarantee low toxin contamination” Which variables?

Considering global warming the dryer and hotter season do not support FHB epidemics in Hungary: for which reason the authors is talking about Hungary in particular?

 In regions like Norther Europe the rate of F. graminearum: please add sensu lato after F.g

“The forecast of Battilani [28] for wheat seems also be real”: the authors should briefly explain the forecast!

Avoiding the use of the personal pronouns throughout the text is highly recommended.

Section 3

“A number of masked and other toxins were identified with very high toxicity [37]”: which toxins?

“In North and Central Europe five Fusarium species were considered as highly pathogenic causing significant epidemics and toxin contamination”: please precise which species are?

“In several parts of China F. asiaticum is dominant (it is also a NIV producer), followed by F. graminearum sensu stricto senso”, please remove the term “senso”, it is called by F. graminearum sensu stricto, sensu stricto should always be written in italic (please check P5)

Table 2: please add the authors of F. cerealis description. F. langsethiae is missing.

Page 5: “F. culmorum, F. avenaceum, F. poae, F. tricinctum, and M. majus (syn. Fusarium) are the dominant species, besides many others”: the species names should be fully written as they are indicated for the first time on the text.

In F. graminearum sensu lato several species are present, but we cannot speak about F.g species complex, F.g belongs to F. sambucinum species complex (FSSC), please also check page 8, page 10...

F. graminearum stricto sensu: please correct the inversion!

“As ruling species may change in time, for example, in North Europe the F. culmorum or F. avenaceum or F. langsethiae (Torp and Nirenberg)”: please remove the brackets.

The authors should add references about the multiple infection (see Senatore et al. 2023).

Section 4:

The authors should precise the total number of samples in each study discussed, this is confusing! Please reformulate this section

“Surprisingly, FB1 was found in 9 samples, FB2 in 8 samples, DON (8), OTA (2), and ZEN (5)”: it is completely normal that other species, producers of FUM, could be present (not causal agents of FHB) such as F. globosum, F. verticillioides and F. proliferatum! It is not surprising!

Section 5:

The authors are shedding the light on other mycotoxins (AFL and OTA) that does not support the main topic of the paper.

“On the other hand, most papers stress the significance of DON reduction in resistance screening but except of about 10-15% of the papers only analyze visual symptoms are measured [7,86,87,88]”: Main DON producers cause important yield losses but there are other species that produce other types of toxins that do not cause losses.

Section 6:

Is the following sentence missing? “Subsequent tests confirmed that the non-specificity of resistance to F. graminearum and F. culmorum correlate very closely; indeed, their resistance is ostensibly the [16,21,92,93]”

“The correlations between species are everywhere close and significant at P = 0.001, with the only exception being F. boothii. This species with F. meridionale has much weaker correlations than the other more aggressive species with the same results”, how many isolates have been analyzed, the non-pathogenicity could be associated to only one isolates, please clarify.

F. acacia-mearnsii, F. asiaticum, and F. cortaderiare belong to F. graminearum sensu lato.

What the following sentence means? When, for example, we have 10 mg/kg DON and 1 mg/kg DON, the NIV data should be multiplied by 10 and summed so that the DON equivalence will be 20 mg/kg” for which reason, I think that this is your own opinion, and I don’t agree with you. I think that the limit dose fixed could be less for instance in case of co-occurrence of multiple toxins in the same sample, but this should be decided by European experts!

Page 11: When the highly resistant plant is ready, the masked DON can be checked: I cannot understand the meanning.

Comments on the Quality of English Language

The english should be checked.

Reviewer 5 Report

Comments and Suggestions for Authors

FHB, caused by different Fusarium species, is one of the most destructive diseases affecting wheat. Heavy epidemics can cause yield losses and toxin contamination. The Fusarium problem in wheat is much more complex than currently supposed. Besides F. graminearum, many other Fusarium species produce harmful toxins that jeopardize food safety. Therefore, it is essential to understand whether FHB resistance has a common resistance to all Fusarium species or whether different genes regulate resistance to different species. In this review, the authors summarized from various points of view the whole mass of the mycotoxin problem of wheat, concentrating more on the food safety effects caused by other Fusarium spp. and their toxins. And the results will be contributed to control the contamination of toxins. The conclusions consistent with the evidence and arguments presented. The text clear and easy to read and can be recommended for publication.

Author Response

Dear Reviewer 5,

I am very thankful for your kind opinion about this manuscript. I hope it helps the Editors to accept the paper. Thank you again your support and your agreement with the content and form of the paper.

Yours truly sincerely